



# Bedrock River Erosion through Dipping Layered Rocks: Quantifying Erodibility through Kinematic Wave Speed

Nate A. Mitchell[1] and Brian J. Yanites[1]

[1]Indiana University, Bloomington, Indiana, USA

*Correspondence to*: Nate A. Mitchell (natemitc@indiana.edu)

**Abstract.** Landscape morphology reflects drivers such as tectonics and climate but is also modulated by underlying rock properties. While geomorphologists may attempt to quantify the influence of rock strength through direct comparisons of
landscape morphology and rock strength metrics, recent work has shown that the contact migration resulting from the presence of mixed lithologies may hinder such an approach. Indeed, this work counterintuitively suggests channel slopes within weaker units can sometimes be higher than channel slopes within stronger units. Here, we expand upon previous work with 1-D stream power numerical models in which we have created a system for quantifying contact migration over time. Although previous studies have developed theory for bedrock rivers incising through layered stratigraphy, we can now scrutinize this theory with
contact migration rates measured in our models. Our results show that previously developed theory is generally robust and that contact migration rates reflect the pattern of kinematic wave speed across the profile. Furthermore, we have developed and tested a new approach for estimating kinematic wave speeds. This approach utilizes stream steepness, a known base level fall rate, and contact dips. Importantly, we demonstrate how this new approach can be combined with previous work to estimate erodibility values. We demonstrate this approach by accurately estimating the erodibility values used in our numerical models.
After this demonstration, we use our approach to estimate erodibility values for a stream near Hanksville, UT. Because we show in our numerical models that one can estimate the erodibility of the unit with lower steepness, the erodibilities we estimate for this stream in Utah are likely representative of mudstone and/or siltstone. The methods we have developed can be applied to streams with temporally constant base level fall, opening new avenues of research within the field of geomorphology.

## 1 Introduction

25       Geomorphologists seek to extract geologic and climatic information from landscape morphology, and the conceptual framework of the stream power model (Howard and Kerby, 1983; Whipple and Tucker, 1999) has driven many such endeavours (Whipple et al., 2013). Indeed, as a representation of bedrock river incision, the stream power model has been used to: (1) identify unrecognized earthquake risks (Kirby et al., 2003); (2) constrain the timing and extent of normal fault activity



(Whittaker et al., 2008; Boulton and Whittaker, 2009; Gallen and Wegmann, 2017); (3) distinguish between potential drivers

of transient incision (Carretier et al., 2006; Gallen et al., 2013; Miller et al., 2013; Yanites et al., 2017); and (4) search for spatial patterns in rock strength (Allen et al., 2013; Bursztyn et al., 2015). This last application is our focus here; to what extent can river morphology be used to detect spatial patterns in rock strength? While such a question seems straightforward to address (e.g., comparing morphologies in different rock types), the mere presence of different rock strengths introduces complicating factors. For example, contact migration perturbs the spatial distribution of erosion rates, causing dramatic

variations in slope along a bedrock river (Forte et al., 2016; Perne et al., 2017; Darling et al., 2020). Surprisingly, these perturbations can even cause streams to counterintuitively have steeper reaches in weaker rocks (Perne et al., 2017). Predicting how contrasts in rock strength are reflected in topography, and specifically river profiles, is therefore necessary to advance our understanding of both (1) the drivers of landscape evolution and (2) how we can use landscape morphology to extract information about geology and climate.

Forte et al. (2016) demonstrated how rock-strength perturbations can cause long-lasting landscape transience, even with constant tectonic and climatic forcing. The spatiotemporal distribution of erosion rates can strongly vary with rock type, with one rock type eroding well above the rock-uplift rate and another well below it. These findings have far-reaching implications for landscape evolution and methodological approaches to quantifying rates. For example, the exposure of a new lithology could trigger an increase in erosion rates, and this increase could be mistakenly attributed to a change in external

forcing such as climate. Variations in erosion rate due to mixed rock types can also influence detrital zircon geochronology, detrital thermochronology, and detrital cosmogenic nuclide analysis (Carretier et al., 2015; Forte et al., 2016; Darling et al., 2020).

As an illustrative example, Fig. 1a shows an area where rivers flow over layered stratigraphy near Hanksville, UT. The streams are underlain by highly variable sedimentary units gently dipping to the west. As the channels flow over these

different units, they exhibit pronounced changes in channel steepness ($k_{sn}$, slope normalized for drainage area; (Wobus et al., 2006)). Indeed, these changes are so dramatic that the profiles have an unusual "stepped" appearance (Figs. 1b and 1c). A logical approach to exploring how these morphologies reflect rock properties would involve (1) quantifying variations in steepness from digital elevation models (DEMs), (2) collecting metrics of rock strength in the field, such as Schmidt Hammer measurements (Murphy et al., 2016) or fracture density (Bursztyn et al., 2015; DiBiase et al., 2018), (3) measuring the

compressive and/or tensile strengths of rock samples taken in the field (Bursztyn et al., 2015), and, finally, (4) exploring patterns between channel steepness and metrics of rock strength. Indeed, the erosion of bedrock rivers through this stratigraphy presents a seemingly convenient opportunity to explore the influence of rock strength. If, however, the migration of contacts along these streams has perturbed the spatial distribution of erosion rates, then identifying relationships between channel steepness and rock strength would be hindered by uncertainties regarding variations in erosion rate. To improve our capacity

to extract rock-strength information from topography, we must understand the influence of such complications.



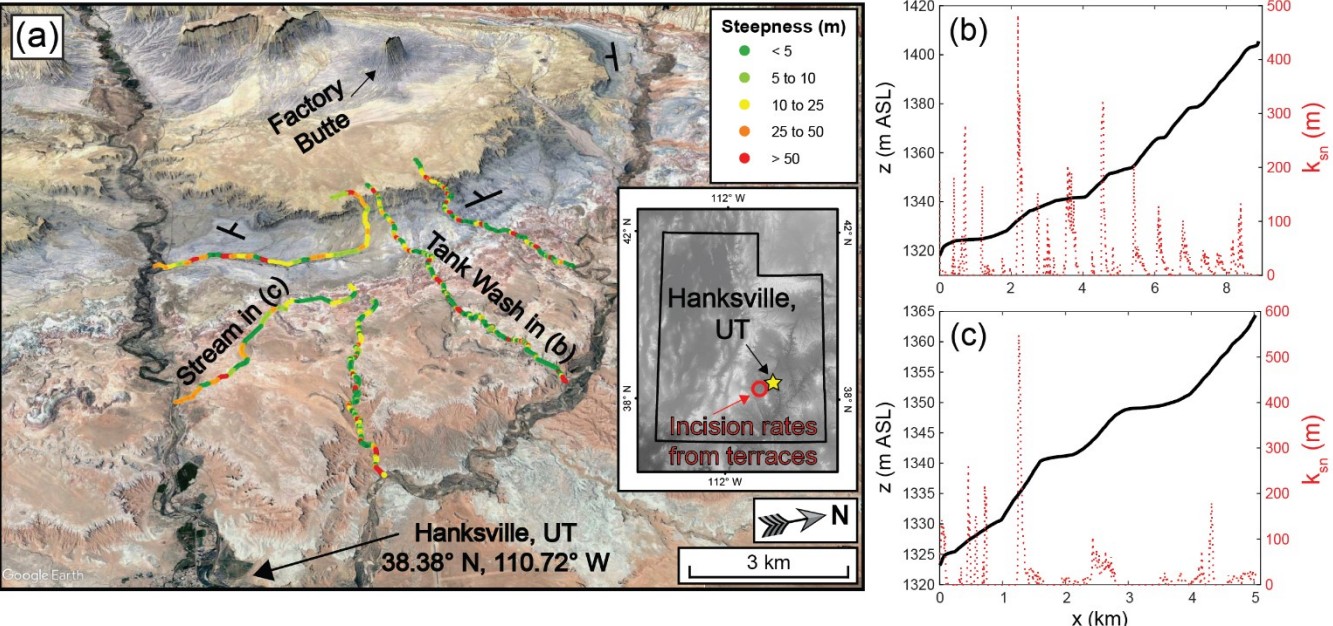

**Figure 1**. Hanksville, UT is a potential example of a landscape where contact migration significantly influences channel morphology. (**a**) Google Earth imagery (© Google Earth) showing variations in stream steepness ($k_{sn}$) near Hanksville. These steepness values were obtained using Topotoolbox v2 (Schwanghart and Kuhn, 2010; Schwanghart and Scherler, 2014). The inset shows the locations of Hanksville, UT (yellow star) as well as the fluvial terraces along the Fremont (red circle) that provide incision rate estimates of 0.3 to 0.85 mm a$^{-1}$ (Repka et al., 1997; Cook et al., 2009). The long profiles of these streams (**b-c**) show strange morphologies that are similar to the conceptual (Fig. 2) and numerical models considered here.

Figure 2 shows several conceptual models for bedrock river incision through layered stratigraphy. We have based these conceptual models on previous work (Perne et al., 2017; Darling et al., 2020) and present them to help guide the reader's understanding of this study's focus. Here, we show only two lithologies (rock type one and rock type two), with these units layered in a repeating succession. Clearly, more complicated situations are possible (e.g., a multitude of lithologies with variable thicknesses, contact dips that vary over space), but our intention here is to present simple examples. The river in Figs. 2a and 2b is incising through horizontal units, and Fig. 2b is at a later time than Fig. 2a. We have shown the river reaches underlain by rock type two to be steeper than those underlain by rock type one. Conventionally, this contrast would lead geomorphologists to suspect that rock type 2 has a higher resistance to erosion than does rock type one. As we discussed above, however, reaches in rock type two may be steeper because they have higher erosion rates (Perne et al., 2017). The stream profile from Fig. 2a is shown in light colors in Fig. 2b, with a red arrow highlighting the fact that the units have been lifted by the rock-uplift rate. This relative uplift could, of course, also represent a falling base level. Although the contacts were raised vertically, the river continues to incise through the units. The positions where the contacts were previously exposed along the river are eroded, causing the contacts to now be exposed further upstream. This balance between ongoing uplift (or base level fall) and river incision causes the contacts to gradually migrate upstream along the stream profile. In Fig. 2c, the units dip in the upstream direction (i.e., a negative contact dip). In this case, the contacts will continue to migrate upstream but the





magnitude of channel slope relative to contact slope varies along the profile. As a result, the influence of contact migration on erosion rates will vary along the profile (Darling et al., 2020). This consequence also applies to the river in Fig. 2d, which is

underlain by units dipping in the downstream direction (i.e., a positive contact dip). Interestingly, when units dip in the downstream direction the contacts can migrate upstream or downstream. The direction of contact migration depends on the contrast between channel slope and contact slope (Darling et al., 2020). Importantly, in each conceptual model shown in Fig. 2 the spatial distributions of erosion rate, steepness, and contact migrate rate all depend on incision model parameters such as erodibility (Perne et al., 2017; Darling et al., 2020). By understanding these landscapes, we may be able to capitalize on these

effects and quantify erodibility in new ways.

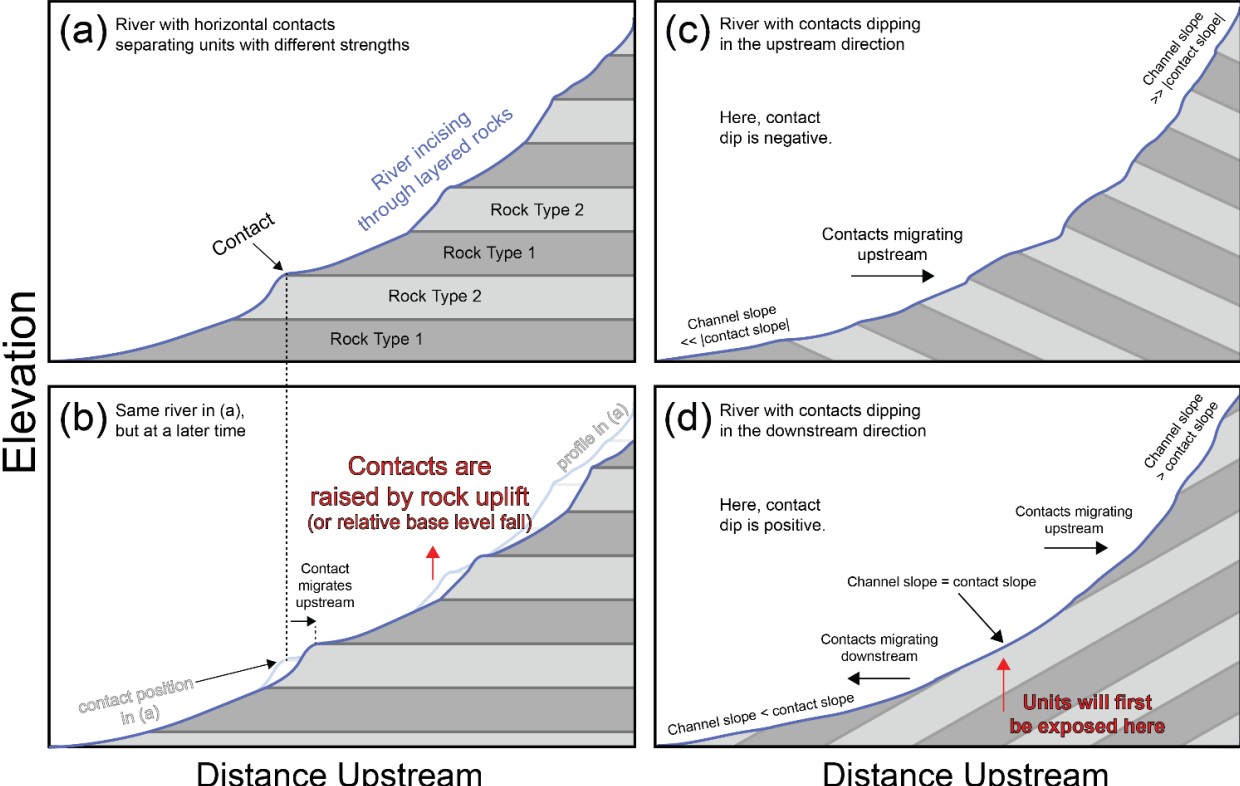

**Figure 2**. Conceptual models of bedrock river incision through layered stratigraphy. Here, only two rock types are shown: rock type 1 and rock type 2. Subplots (**a-b**) show a river with a contact dip ($\phi$) of 0°. Subplot (**b**) is at a later time than subplot (**a**) to demonstrate how the contacts gradually migrate upstream along the stream profile. Subplot (**c**) shows a stream with

contacts dipping in the upstream direction ($\phi < 0°$). Subplot (**d**) shows a stream with contacts dipping in the downstream direction ($\phi > 0°$).

Here, we focus on how to use contact migration to gain insight into bedrock river behavior and quantify erodibility. Rather than being a complication to be avoided, the perturbation introduced by contact migration can be exploited for model calibration in a similar manner to how one would exploit a transient response to tectonics (Whipple, 2004). Although previous

work has developed theory for bedrock river erosion and steepness when contact dips are zero (Perne et al., 2017) or nonzero (Darling et al., 2020), this theory has not been thoroughly compared with observations from numerical models. Here, we use





numerical models in which we can measure contact migration over time to pursue the following research questions: (1) Does the theory developed by Perne et al. (2017) for river incision through horizontal strata accurately reflect observations from numerical models? (2) Does the theory developed by Darling et al. (2020) for river incision through nonhorizontal strata
accurately reflect observations from numerical models? (3) What is the potential for using these theoretical frameworks to estimate incision model parameters like erodibility for real bedrock rivers? By developing a new method for estimating kinematic wave speed, we will show that morphologic metrics like steepness and contact dip can be used to estimate bedrock erodibility, even where contact dips are nonzero. The dynamics of landscapes with layered rocks are increasingly shown to be quite rich (Glade et al., 2017; Ward, 2019; Sheehan and Ward, 2020, 2020b), and these landscapes offer valuable opportunities
to compare expectations shaped by model results with the unflinching testimony of the field. Our intention here is to (1) further elucidate what we should expect from the common form of the stream power model and (2) provide a framework for quantifying rock strength from bedrock river morphology. After developing this framework through the use of numerical models, we demonstrate its application on Tank Wash, one of the streams near Hanksville, UT (Fig. 1).

## 2 Methods

115       To address the research questions outlined above, we model one-dimensional river profiles using the stream power equation. We set up a series of model experiments to expand upon previous work and generate a framework for extracting quantitative information about erodibility and stream dynamics from profile morphology in areas with complex lithology. Table 1 summarizes the numerical model scenarios we explore. We detail these models further below, but at this point we emphasize that we examine four distinct model scenarios: (1) models with two rock types and horizontal contacts; (2) models
with three rock types and horizontal contacts; (3) models with two rock types and contacts dipping in the upstream direction; and (4) models with two rock types and contacts dipping in the downstream direction. We use the first two model scenarios to test and further explore the framework developed by Perne et al. (2017) (i.e., bedrock river incision through layered rocks when the contact dip is zero), and we use the last two model scenarios to test and further explore the framework developed by Darling et al. (2020) (i.e., bedrock river incision through layered rocks when contact dips are nonzero).

**Table 1.** Parameter space of the four model scenarios.

| Scenario | Description | $U$ (m a$^{-1}$) | $H$ (m) | $\phi$ (°) | $n$ | $m$ | Reference $K_W$ (m$^{1-2n\theta}$ a$^{-1}$) † |
|---|---|---|---|---|---|---|---|
| 1 | Two layers, zero dip | 0.15 m a$^{-1}$ and 0.3 m a$^{-1}$ | (1) $H_W = H_S = 100$ m, (2) $H_W = H_S = 150$ m, (3) $H_W = 75$ and $H_S = 150$ m, and (4) $H_W = 150$ and $H_S = 75$ m | 0° | 0.67 and 1.5 | n × 0.5 | For n = 0.67: (1) 4.29×10$^{-6}$, (2) 6.83×10$^{-6}$, and (3) 1.09×10$^{-5}$. For n = 1.5: (1) 1.57×10$^{-8}$, (2) 4.44×10$^{-8}$, and (3) 1.26×10$^{-7}$ |
| 2 | Three layers, zero dip | 0.15 m a$^{-1}$ | All $H = 100$ m | 0° | 0.67 and 1.5 | n × 0.5 | |
| 3 | Two layers, dipping upstream | 0.15 m a$^{-1}$ | All $H = 100$ m | -0.5°, -1°, -2.5°, -5°, -10°, -15°, -25°, -35°, and -45° | 0.67 and 1.5 | n × 0.5 | |
| 4 | Two layers, dipping downstream | 0.15 m a$^{-1}$ | All $H = 100$ m | 0.5°, 1°, 2.5°, 5°, 10°, and 15° | 0.67 and 1.5 | n × 0.5 | |

† The erodibilities of the medium layer ($K_M$, only present in scenario 2) and strong layer ($K_S$, present in all scenarios) are set relative to the reference weak erodibility ($K_W$), as discussed in Sect. 2.2.



## 2.1 Bedrock River Erosion and Morphology

In this section, we present the basics of bedrock river erosion and the morphologic metrics we use to study it. We use
a first-order upwind finite difference scheme to represent the stream power model (Howard and Kerby, 1983; Whipple and Tucker, 1999):

$$\frac{\delta z}{\delta t} = U - E = U - KA^m \left| \frac{\delta z}{\delta x} \right|^n \tag{1}$$

where $z$ is elevation (L), $t$ is time (T), $U$ is rock-uplift rate (L T$^{-1}$), $E$ is erosion rate (L T$^{-1}$), $K$ is erodibility (L$^{1-2m}$ T$^{-1}$), $A$ is drainage area (L$^2$), $x$ is distance upstream (L), and both $m$ and $n$ are exponents. These exponents reflect erosion physics and
the scaling of both channel width and discharge with drainage area (Whipple and Tucker, 1999; Lague, 2014). The ratio of $m/n$ has been shown to influence river concavity ($\theta$) at steady state and uniform rock-uplift and erodibility (Tucker and Whipple, 2002). We use $m/n = 0.5$, which falls within the expected range of $m/n$ values (Whipple and Tucker, 1999) and is consistent with many other studies (Farías et al., 2008; Gasparini and Whipple, 2014; Han et al., 2014; Mitchell and Yanites, 2019). Because slope exponent $n$ strongly influences bedrock river dynamics (Tucker and Whipple, 2002), we evaluate $n$
values of 0.67 and 1.5. Although $n$ is often assumed to equal one (Farías et al., 2008; Fox et al., 2014; Goren et al., 2014; Ma et al., 2020), we explain in Sect. 2.4 why we do not evaluate models with $n = 1$.

For an equilibrated stream ($dz / dt = 0$) with uniform properties, channel steepness $k_{sn}$ is related to the ratio of rock-uplift rates to erodibility (Hack, 1973; Flint, 1974; Duvall et al., 2004; Wobus et al., 2006):

$$k_{sn} = \left| \frac{dz}{dx} \right| A^{m/n} = \left( \frac{U}{K} \right)^{1/n} \tag{2}$$

Equation 2 has shaped the focus of many studies in tectonic geomorphology (Wobus et al., 2006). Although this framework is powerful, the streams we examine here have spatially variable properties (i.e., $K = f(x)$). This distinction will cause variations in channel slope and steepness that are not captured by Eq. 2, and we seek to further understand these variations in slope and steepness.

We use Hack's Law (Hack, 1957) to set each river's drainage area:
$$A(x) = C(\ell - x)^h \tag{3}$$

where x is distance upstream from the stream's outlet, $\ell$ is the length of the drainage basin (taken as 20.6 km), $C$ is a coefficient (L$^{2-h}$), and $h$ is an exponent. We use $C$ and $h$ values of 1 m$^{0.2}$ and 1.8, respectively. All streams are 20 km long, so using $\ell = 20.6$ km makes the rivers have a maximum drainage area of about 58 km$^2$. This $\ell$ value also causes the critical drainage area (0.1 km$^2$) to occur where $x = 20$ km. We use a distance between stream nodes of $dx = 5$ m.

We present the resulting stream profiles as $\chi$-plots here because $\chi$-$z$ space removes the influence of drainage area on channel slope (Perron and Royden, 2013; Mudd et al., 2014):

$$\chi = \int_{x_b}^{x} \left( \frac{A_0}{A(x)} \right)^{m/n} dx \tag{4}$$

where $\chi$ is transformed distance upstream (L), $x_b$ is the position of base level ($x = 0$ m), and $A_0$ is a reference drainage area (here, taken as 1 km$^2$).





An effective method to compare channel slopes and contact dip $\phi$ is to use the slope of the contact in $\chi$-space, which we refer to as $\phi_\chi$:

$$\phi_\chi = \frac{dz_{contact}}{d\chi} \qquad (5)$$

where $z_{contact}$ is contact elevation (L) and $\chi$ is that of the stream node directly above the contact position in question. Admittedly, comparing contact elevations with $\chi$ may be initially confusing, as $\chi$ is related to river elevations rather than contact elevations.

Utilizing the apparent contact dip in $\chi$-space is advantageous, however, because it encapsulates the influence of both drainage area and contact dip in real space. If one decides to utilize only drainage area or contact dip, then the influence of the excluded metric would not be present in one's analyses. Note that we will present $\phi_\chi$ as dimensionless values (i.e., the change in elevation over the change in transformed river distance), while we present contact dip $\phi$ in degrees.

**2.2 Defining the range of erodibility values**

The contrast in erodibility ($K$) values between weak and strong layers is one of the most important controls on bedrock river incision through layered stratigraphy, and we therefore explore this parameter space thoroughly. Selecting $K$ values for different simulations is not a simple matter, however. The way erodibility influences river dynamics depends on the exponents $m$ and $n$, so the effects of a two-fold difference in $K$ on both stream morphology and erosion dynamics is not the same for $n = 0.67$ and $n = 1.5$. Furthermore, comparing $K$ values is context dependent. For example, $K$ values could be selected to either (1)

provide a similar range of channel elevations (Beeson and McCoy, 2020) or (2) allow similar timescales for transient adjustment (Mitchell and Yanites, 2019). Oftentimes, one cannot fulfill multiple such requirements when selecting erodibilities and one must choose a specific approach. Because we examine different $n$ values here, we set the range of erodibility by considering slope patch migration rates (Royden and Perron, 2013):

$$\frac{d\chi_{sp}}{dt} = \frac{nU}{\left(\frac{U}{K}\right)^{1/n} A_0^{-m/n}} \qquad (6)$$

where $\chi_{sp}$ is the $\chi$ value of a slope patch and $U$ is the rock-uplift rate (or base-level fall rate) at the time $t$ when the slope patch was generated at the basin's outlet (constant over time here; $U \neq f(t)$). We vary $K$ values so that slope patch migration rates calculated for the stronger layer are a certain fraction (33, 50, 67, 75, or 90 %) of those calculated for the weaker layer (using $U$ for both slope patch migration rates; Eq. 6). Although these slope patch migration rates are calculated with $U$, these rivers do not always erode at $U$. The rock-uplift rate is still a physically representative rate, however (i.e., erosion rates will vary around $U$). Most importantly, this approach to scaling erodibility is consistent across different $n$ scenarios.

The erodibility for the weak layer ($K_W$) is set to a reference value (Table 1), and the erodibility of the strong layer ($K_S$) is set according to the modelled scenario (e.g., the percentages listed in the previous paragraph). We use three reference weak erodibilities ($K_W$) for each $n$ value. For $n = 0.67$, the low, moderate, and high erodibilities for the weak layer are $4.29 \times 10^{-6}$, $6.83 \times 10^{-6}$ and $1.09 \times 10^{-5}$ m$^{0.33}$ a$^{-1}$. For $n = 1.5$, the low, moderate, and high erodibilities for the weak layer are $1.57 \times 10^{-8}$, $4.44 \times 10^{-6}$ and $1.26 \times 10^{-7}$ m$^{-0.5}$ a$^{-1}$. We chose the low, medium, and high reference $K_W$ values so that they allow slope patch

migration rates equal to those for $n = 1$, $U = 0.15$ mm a$^{-1}$, and $K$ values of $5 \times 10^{-7}$ a$^{-1}$, $10^{-6}$ a$^{-1}$, and $2 \times 10^{-6}$ a$^{-1}$, respectively.



In the simulations using three rock types (scenario 2; Table 1), the erodibilities for the medium unit are scaled to be between those for the weak and strong units. For each combination of weak and strong erodibilities ($K_W$ and $K_S$), three erodibilities are evaluated for the medium rock type ($K_M$): (1) the $K_M$ that provides the average of the slope patch migration rates in the strong and weak units (for rock-uplift rate $U$); (2) the $K_M$ that provides a slope patch migration rate that is three times closer to that of the strong unit (for rock-uplift rate $U$); and (3) the $K_M$ that provides a slope patch migration rate that is three times closer to that of the weak unit (for rock-uplift rate $U$). For example, if the $K_S$ value provides slope patch migration rates that are 50 % of those in the weak unit, $K_M$ would be varied to provide slope patch migration rates that are 62.5, 75, and 87.5 % of those in the weak unit for three separate model simulations.

## 2.3 Recording contact migration rates

We track and record contact positions over time in our simulations. We record each contact's position every 25 ka, which is larger than model time step $dt = 25$ a. Contact migration rates are recorded for a total of 10 Myr for each simulation. Note that before we begin recording contact migration rates, we run each simulation for a time period sufficient to allow the range of river elevations to become constant over time (i.e., a state of dynamic equilibrium). When $n < 1$, we initialize the river elevations using the steepness (Eq. 2) for steady conditions and the strong layer's erodibility ($k_{sn} = (U / K_S)^{1/n}$). When $n > 1$, we initialize the river elevations using the steepness for steady conditions and the weak layer's erodibility ($k_{sn} = (U / K_W)^{1/n}$). We based these choices on observations regarding how average river steepness varies for different $n$ values. After we initialize the rivers elevations, the rivers need some time to adjust from the initial conditions. Although the rivers quickly arrive at morphologies like those shown in our conceptual model (Fig. 2), the river elevations can gradually increase or decrease before finally arriving upon a consistent range of elevations. Streams in scenarios 1, 2, and 4 (Table 1) were given 50 Myr to adjust, while streams in scenario 3 were given 100 Myr to adjust. These adjustment times ensured that the streams in all simulations had achieved a dynamic equilibrium (i.e., the range of elevations became constant with time; Figs. S1-S4).

The aspect of the contact tracking system we stress here is that recording contact migration rates near the basin outlet is inadvisable because of base-level perturbations (Perne et al., 2017). We describe how we avoid the problems caused by base level perturbations further below, but first we provide some background on the base-level perturbations themselves. Previous work (Forte et al., 2016; Perne et al., 2017; Darling et al., 2020) shows that contact migration dynamics can cause much of the profile to have erosion rates that are not equal to the rock-uplift rate ($U$). Near the outlet, however, the constant forcing of the base level fall rate at this hard boundary means that an equilibrated reach eroding at $U$ is constantly being created at the basin outlet. This reach with $E = U$ is at odds with the rest of the profile, where $E$ is instead a function of contact migration dynamics. As the units are steadily raised by rock uplift, the equilibrated reach near the outlet is perturbed when a different rock type is exposed beneath it. For example, a reach in the strong unit may be connected to the outlet (causing it to have $E = U$), only for the weak unit to appear beneath it. Then, an equilibrated reach in the weak unit with $E = U$ begins to form, and the previously equilibrated reach in the strong unit is disconnected from the basin outlet. The erosion at the base of the previously equilibrated reach in the strong unit will then change in order to conform with the dynamics of the contact's migration. This change in erosion rate can, in certain situations, create a consuming knickpoint (Royden and Perron, 2013) that migrates upstream. These





knickpoints are the reason why recording contact positions near the basin outlet is problematic for our comparison of measured contact migration rates and estimated kinematic wave speeds; one would record pronounced variations in contact migration rates near the outlet as reaches initially have $E = U$ and are then perturbed by knickpoints once they are disconnected from the basin outlet. To test the equations for kinematic wave speed we examine here, we must examine these equations where they are applicable (i.e., not where knickpoints from base level effects dominate). The expression of these knickpoints will, however, sufficiently diminish at some distance upstream from the basin outlet. To characterize the upstream extent of these knickpoints, Perne et al. (2017) derived the damping length scale $\lambda$. This damping length scale defines the $\chi$ distance from the outlet at which the effects of such disturbances have largely disappeared:

$$\lambda = H_S \left(\frac{K_S A_0{}^m}{U}\right)^{1/n} \left[1 + \left(\left(\frac{K_W}{K_S}\right)^{1/n} - 1\right)^{-1}\right] \tag{7}$$

where $H_S$ is the layer thickness of the stronger unit. Note that $\lambda$ is calculated as the $\chi$-distance from the outlet at which a knickpoint migrates from the bottom to the top of a reach underlain by the strong layer (Perne et al., 2017). We will display the position where $\chi = \lambda$ in our figures, and we also highlight an example of the consuming knickpoints that can form in the manner described above.

We avoid such base-level perturbations by only recording and analyzing contact migration rates upstream of the location where $\chi = \lambda$. Depending on the parameters used in a simulation, however, $\lambda$ can sometimes be larger than the highest $\chi$ value. Otherwise, if $\lambda$ is too close to the highest $\chi$ value, this may leave only a few stream nodes for analysis. Tracking contact migration over such a small portion of the stream profile can cause issues for our approach (e.g., a reach extending across one rock type may not fit within the area). We therefore only record contact migration if a minimum proportion of the profile can be used (25 % of the profile for scenarios 1-3 and 10 % of the profile for scenario 4; these choices were based on model observations). We do not utilize results for simulations that do not meet these criteria, but this loss is alleviated by the large number of simulations we assess.

Quantifying contact migration in some of our numerical models requires additional considerations, however. Such models include (1) those with three rock types instead of two and (2) those where contacts dip downstream. When we evaluate numerical models with three rock types (weak, medium, and strong), the issue is that damping length scale $\lambda$ is calculated using the parameters for only two rock types. When we use three rock types, we therefore calculate $\lambda$ between the weak and medium layers and between the medium and strong layers. We use the larger of these two $\lambda$ values to set the starting position for contact tracking. Although this approach for simulations with three rock types is simple, there are far more considerations for numerical models with contacts dipping in the downstream direction. For example, the conceptual model in Fig. 2d has contacts dipping in the downstream direction. Based on previous work (Darling et al., 2020), we anticipate that locations where channel slope is less than the contact slope will have contacts that migrate in the downstream direction. We do not focus on contact migration in the downstream direction here; when contacts migrate in the downstream direction, they migrate in the opposite direction of base level signals. As a result, these contacts (1) do not have the same capacity for driving feedbacks between erosion and contact migration and (2) do not offer the same opportunities for extracting information regarding rock strength in the manner



we propose in this study. Downstream-migrating contacts may still offer other kinds of valuable information, but we restrict
this study's focus to upstream-migrating contacts. When contacts dip downstream, our approach is to only record contact
migration rates upstream of the point where contact slope is exceeded by the average channel slope (calculated with the
profile's average steepness, drainage area, and $m/n$; Eq. 2) exceeds the contact slope. We use the average steepness to find this
point because channel slope at each point can change considerably, while the profile's average steepness does not. The exact
location where new units are exposed along the profile varies, however, so the starting position must be advanced some distance
upstream (otherwise, we would record downstream-migrating contacts). We discuss these considerations in the supplement.
After moving the starting position for contact tracking some distance upstream from the point where average channel slope
equals the contact slope, we call the $\chi$ value at this location $\zeta$ (Eq. S2). In scenario 4 ($\phi > 0°$; Table 1), we record contact
migration rates upstream of the location where $\chi = \zeta$ because the damping length scale $\lambda$ (Eq. 7) is not always applicable.

**2.4 Erosion and Kinematic Wave Speed for Horizontal Units**

Now, we delve further into (1) the erosion rate variations that occur during river incision through layered stratigraphy
and (2) how these erosion rate variations influence kinematic wave speed. We will show that kinematic wave speed is an
important concept for this research because it is closely linked to contact migration along rivers. In this section, we review the
semi-analytical framework developed for kinematic wave speed when contact dip is zero (Perne et al., 2017).

         Before we outline the semi-analytical framework developed in previous work, we provide a background on how
erosion rate relates to kinematic wave speed. Contact migration can cause the erosion rates on either side of a contact to change
(Perne et al., 2017; Darling et al., 2020). One side of the contact can have an erosion rate below the base level fall rate, while
the other side can have an erosion rate above it. These erosion rate variations occur so that both sides of the contact have the
same horizontal retreat rate in the upstream direction. This retreat rate is closely related to the concept of kinematic wave speed
($C_H$) (Rosenbloom and Anderson, 1994):

$$C_H = KA^m \left|\frac{dz}{dx}\right|^{n-1} = KA^{m/n} k_{sn}^{n-1} = KA^{m/n} \left(\frac{E}{K}\right)^{\frac{n-1}{n}}$$
(8)

Note that Eq. 8 suggests $C_H$ has a power-law relationship with drainage area ($A$). Although this is the general equation for
kinematic wave speed, the challenge when considering rivers incising through layered rocks is what erosion rate $E$ is
appropriate. Kinematic wave speed can be regarded as the migration rate of signals along rivers. When considering such
signals, geomorphologists usually think of base level fall due to tectonic activity (e.g., normal faulting) or drainage capture.
The signals we are concerned with here, however, are the erosional signals arising from contact migration. Because the
exposure of a new rock type can perturb erosion rates (Forte et al., 2016), even without changes in external drivers like base
level fall rate and climate, contact migration is an autogenic perturbation. As erosion causes a contact to migrate upstream
along a river, the autogenic signal persists and becomes a significant influence on river morphology.

         Perne et al. (2017) showed that when contacts are horizontal ($\phi = 0°$), river reaches underlain by weak and strong rock
types will have characteristic steepness values that reflect the layers' relative difference in erodibility. Surprisingly, the steeper
reaches can sometimes be within the weaker rock type. Whether reaches in the strong or weak rock type are steeper depends





on $n$, the parameter that controls how erosion rate scales with channel slope (Eq. 1). Slope exponent $n$ plays an important role in bedrock river incision through layered stratigraphy because it controls how kinematic wave speed scales with erosion rate (Eq. 8). When $n > 1$, $C_H$ is directly proportional to $E$. When $n < 1$, $C_H$ is inversely proportional to $E$. And when $n = 1$, $C_H$ is

independent of $E$. Strangely, this insensitivity of $C_H$ to $E$ when $n = 1$ causes channels incising through layered rocks to consist only of flat reaches and vertical steps (Fig. S5). The channel slopes when $n = 1$ are either infinite or zero (infinite at the steps and zero in the flat reaches). Although this morphology has been compared with waterfalls (Perne et al., 2017), waterfall dynamics are quite distinct (Lamb et al., 2007; Haviv et al., 2010; Scheingross and Lamb, 2017) and require a different treatment. We do not intend to explicitly portray waterfalls here, and we therefore focus on models with $n$ values of 0.67 and

1.5. When $n < 1$, the weaker rock type has higher channel slopes and erosion rates (Perne et al., 2017). Conversely, when $n > 1$ the strong rock type has higher channel slopes and erosion rates. We will use observations from our numerical models to further demonstrate why these behaviors emerge.

Because river incision through layered rocks in highly dependent on slope exponent $n$, simply evaluating the ratio of the weak and strong layers' erodibilities ($K_W / K_S$) is not an effective way to encapsulate the influence of contact migration.

Instead, we use a term that is a function of the weak and strong erodibilities as well as slope exponent $n$. Darling et al. (2020) showed that if you consider weak and strong rock types (with kinematic wave speeds $C_{HW}$ and $C_{HS}$, erodibilities $K_W$ and $K_S$, steepness values $k_{snW}$ and $k_{snS}$, and erosion rates $E_W$ and $E_S$, where the subscripts refer to weak and strong layers, respectively) and set the kinematic wave speeds equal to one another, you arrive at an equation Perne et al. (2017) derived through a different approach. Because many readers will be unfamiliar with this work, we show the derivation in three parts.

$$C_{HW} = K_W\,A^{m/n}\,k_{snW}{}^{n-1} = C_{HS} = K_S\,A^{m/n}\,k_{snS}{}^{n-1} \tag{9a}$$

$$\frac{K_W}{K_S} = \left(\frac{k_{snS}}{k_{snW}}\right)^{n-1} \tag{9b}$$

$$\frac{k_{snW}}{k_{snS}} = \left(\frac{K_S}{K_W}\right)^{\frac{1}{n-1}} = \left(\frac{K_W}{K_S}\right)^{\frac{1}{1-n}} = K^* \tag{9c}$$

We refer to this ratio as $K^*$. Note that if you express $k_{snW}$ and $k_{snS}$ in Eq. 9c using Eq. 2 (e.g., $k_{snW} = (E_W / K_W)^{1/n}$), you will also find that:

$$K^* = \frac{E_W}{E_S} \tag{10}$$

Where $E_W$ and $E_S$ are the erosion rates in the weak and strong layers, respectively. Even though $K^*$ represents the contrast in erosion or steepness between weak and strong rock types when the contact dip is zero, we will show that $K^*$ is still an effective metric for erodibility contrasts when contact dips are nonzero (in the form $(K_W / K_S)^{1/(1-n)}$). Although Darling et al. (2020) derived $K^*$ by considering kinematic wave speeds, Perne et al. (2017) derived $K^*$ by assuming that:

$$\frac{E_W}{E_S} = \frac{\left(k_{snW}\,A^{-m/n}\right) - \tan(\phi)}{\left(k_{snS}\,A^{-m/n}\right) - \tan(\phi)} \tag{11}$$

where $\phi$ is the contact dip in degrees (positive when dipping in the downstream direction). The concept behind this approach is that if the contact dip is high (e.g., vertical contacts that do not migrate horizontally), the right side of Eq. 9 approaches one.



In that case, rearranging the equation would return us to the general expectations formed without considering contact migration: that the erosion rate is the same within each rock type ($E_W = K_W A^m |dz/dx|^n = E_S = K_S A^m |dz/dx|^n$). If, however, the channel slopes are much higher than the contact slopes, then $\phi$ can be considered to go to zero. If $\phi = 0°$, replacing the erosion rates in Eq. 11 with the stream power equation (Eq. 1) and rearranging leads to $K^*$ (Eq. 9c). A similar approach was used by Imaizumi et al. (2015), albeit for the retreat of rock slopes rather than rivers.

The semi-analytical framework presented by Perne et al. (2017) can be used to calculate kinematic wave speeds for bedrock rivers incising through horizontal strata. We use measurements from our numerical models to test the accuracy of predictions made with their framework. To calculate the kinematic wave speed for a reach underlain by a strong layer, you must first solve for the erosion rate as (Perne et al., 2017):

$$E_S = U \frac{\left(\frac{H_S}{H_W}\right) + \left(\frac{K_S}{K_W}\right)^{\frac{n}{1-n}+1}}{1 + \left(\frac{H_S}{H_W}\right)} \tag{12}$$

where $E_S$ is the erosion rate in the stronger layer (L T$^{-1}$) and $H_S$ and $H_W$ are the layer thicknesses (L) of the strong and weak layers, respectively. To calculate the weak erosion rate ($E_W$) for a contact dip of zero, the strong and weak indices in Eq. 12 can simply be reversed. The kinematic wave speed within one layer can then be estimated by inserting Eq. 12 into the general equation for kinematic wave speed (Eq. 8). Note that Perne et al. (2017) derived Eq. 12 by assuming that the average erosion at each point along the stream, over time, must balance the rock-uplift rate ($U$). To understand the concept behind this approach, first consider a reach that is underlain by one rock type and eroding at a rate above $U$. As erosion causes the contacts defining that reach to migrate upstream over time, the reach creates an imbalance between the river's erosion rate and rock-uplift rate. After the reach migrates past one position, however, its migration is followed by a reach in another rock type. This rock type would have an erosion rate below the rock-uplift rate, and the passage of this low-erosion reach restores the balance over time between rock uplift and erosion. Perne et al. (2017) based this perspective on observations from their numerical models; despite oscillations in channel slope as contacts migrate upstream, the rivers reached a dynamic equilibrium such that the range of elevations was constant over time. This dynamic equilibrium suggests that erosion and rock uplift do balance each other, over a sufficient time interval (i.e., the time for reaches in both rock types to migrate past a position). Given the assumptions involved in the derivation of Eq. 12, however, we will use measurements from our numerical models to test its accuracy.

## 2.5 Erosion and Kinematic Wave Speed for Nonhorizontal Units

When units have nonzero dips, the dynamics between erosion rate and kinematic wave speed change entirely (Darling et al., 2020). To evaluate how bedrock river erosion rates vary with contact dip, we fit multilinear regressions to our model results in the form:

$$\frac{E_W}{U} = f\left(K^*, \ln\left(|\phi_\chi|\right)\right) \tag{13}$$

where $E_W / U$ is the average erosion rate of the weak unit normalized by rock-uplift rate, $K^*$ is a metric for the erodibility contrasts between weak and strong layers (Eq. 9c), and $\phi_\chi$ is the contact dip in $\chi$-space (Eq. 5). The purpose of this approach is to demonstrate how erosion rates change with drainage area, contact dip (both of which influence $\phi_\chi$), and contrasts in rock





strength. We take the average $E_W / U$ and $\ln(|\phi_\chi|)$ values within ten drainage area bins spaced logarithmically from the highest to lowest drainage areas. Utilizing the logarithm of $|\phi_\chi|$ is effective because this drainage-area proxy aids in portraying the power-law relationships surrounding drainage area in the stream power model (Eq. 1). Excluding the influence of contact dip by using drainage area instead of $\ln(|\phi_\chi|)$ would, for example, provide only scatter rather than the three-dimensional relationships we will demonstrate between $E_W / U$, $K^*$, and $\ln(|\phi_\chi|)$. For these analyses, we only use erosion rates from the final

model time step (rather than values over the entire 10 Myr duration).

Now, we present the framework for kinematic wave speeds along bedrock rivers incising through nonhorizontal strata. Darling et al. (2020) used geometric considerations to solve for the kinematic wave speed as:

$$C_H = \frac{K_W A^m \left|\frac{dz}{dx}\right|^n}{\left|\frac{dz}{dx}\right| - \tan(\phi)} = \frac{A^m U}{\left(\left(\frac{U}{K_W}\right)^{1/n} A^{-m/n}\right) - \tan(\phi)} \tag{14}$$

where the weak layer is assumed to erode at rock-uplift rate $U$. Note that Eq. 14 suggests that when $\phi < 0°$ (dipping upstream),

$C_H$ will be lower (i.e., the denominator will increase). When $\phi > 0°$ (dipping downstream), $C_H$ will be higher (i.e., the denominator will decrease). Although $C_H$ usually increases as a power-law function of drainage area (Eq. 8), Eq. 14 also suggests that nonzero contact dips will cause a departure from the power-law relationships typically expected (i.e., in a log-log plot of $C_H$ vs drainage area, the data will no longer follow a linear trend). While Perne et al. (2017) showed that the erosion rate of the weak layer changes when contact dip is zero, Darling et al. (2020) assumed that the weak layer erodes at the base

level fall rate. Darling et al. (2020) focused on scenarios with $n > 1$; because the strong layer is the less steep layer when $n < 1$, we will use the parameters of the strong layer ($K_S$) in Eq. 14 when $n < 1$.

In addition to Eq. 14, we present an alternative method to estimate kinematic wave speed from channel steepness. We have essentially modified the approach of Darling et al. (2020) to utilize observed channel steepness. The approach remains applicable whether the contact dip is zero or nonzero, and although it utilizes a base level rate ($U$) it is not based on assumptions

regarding the erosion rate within each layer. Kinematic wave speed $C_H$ for a reach underlain by one rock type can be estimated as:

$$C_H = \frac{\left(\frac{U}{k_{sn}{}^n}\right) A^m \left(k_{sn} A^{-m/n}\right)^n}{\left(k_{sn} A^{-m/n}\right)^n - \tan(\phi)} = \frac{U}{\left(k_{sn} A^{-m/n}\right)^n - \tan(\phi)} \tag{15}$$

where $k_{sn}$ is the average steepness observed for the reach spanning the layer in question. We estimate the $K$ value in Eq. 15 as $U / k_{sn}{}^n$; this approach assumes the reach is equilibrated to $U$ and previous work (Forte et al., 2016; Perne et al., 2017; Darling

et al., 2020) suggests that assumption can be incorrect. The advantage of this approach, however, lies in taking the average of Eq. 15 estimates from multiple rock types. For example, the Eq. 15 $C_H$ estimates for one rock type will be too high, while the $C_H$ estimates for the other rock type will be too low. By taking the average of both estimates, the deviations in erosion rate balance each other out and provide an accurate estimate of $C_H$. Importantly, this approach can then be combined with that of Darling et al. (2020) (Eq. 14). By utilizing Eqs. 14 and 15 together, one can compare $C_H$ estimates based only on quantifiable

metrics ($U$, $k_{sn}$, and $\phi$ in Eq. 15) and $C_H$ estimates calculated using a specified erodibility (Eq. 14). We will show that this





combination can allow the estimation of erodibility for real streams, like those near Hanksville, UT (Fig. 1). We describe this combination further below, after we provide more details regarding the application of Eq. 15.

To make estimates of $C_H$ with Eq. 15, we utilize the following procedure: (1) create bins defined by drainage area values, with ten bins spaced logarithmically from the lowest to the highest drainage areas; (2) for each drainage area bin, take

the average steepness ($k_{sn}$) within each rock type; (3) using the average steepness for each rock type, calculate $C_H$ with Eq. 15; and (4) take the average of the $C_H$ estimates from both rock types in each drainage area bin. Note that this approach requires an independent estimate of rock-uplift rate $U$ (or base-level fall rate) as well as contact dip $\phi$. Due to the data limitations for real streams, we will compare contact migration rates measured in our models with Eq. 15 $C_H$ estimates that use only $k_{sn}$ values from the final model timestep of each simulation. We show Eq. 15 estimates using the entire 10 Myr of recorded $k_{sn}$ values for

each simulation in the supplement, however. Equation 14 does not use $k_{sn}$ values recorded over time.

Now, we describe how we combine the framework developed by Darling et al. (2020) (Eq. 14) with our approach (Eq. 15) in the evaluation of our numerical models. Note that we perform this comparison to test how well it can recover erodibility values from river morphology in a numerical model; establishing this accuracy is important because we apply similar analyses to Tank Wash near Hanksville, UT (Fig. 1). We describe our analysis of Tank Wash in Sect. 2.6 below. To

test the effectiveness of this approach in the numerical models, we compare the average Eq. 15 $C_H$ estimate within each drainage area bin with Eq. 14 $C_H$ values calculated using the enforced contact dip ($\phi$) and slope exponent ($n$) and a wide range of erodibilities ($K$, 200 values spaced logarithmically from $10^{-9}$ to $10^{-4}$ m$^{1-2n\theta}$ a$^{-1}$, where $\theta = 0.5$). We compare the two sets of kinematic wave speed estimates with the X$^2$ Misfit Function (Jeffery et al., 2013):

$$X^2 = \frac{1}{N-\nu-1}\Sigma_i^N \left(\frac{sim_i - obs_i}{tolerance}\right)^2 \qquad (16)$$

where $N$ is the number of observations being compared (up to ten, for average values in the ten drainage area bins), $\nu$ is the number of free variables ($\nu = 1$ here, as we only vary $K$ in Eq. 14), $sim_i$ and $obs_i$ are the $C_H$ estimates from Eq. 14 and Eq. 15 in each drainage area bin, respectively, and $tolerance$ is taken as 1 m a$^{-1}$. Note that we take the $obs_i$ values as the average $C_H$ estimates made with Eq. 15; we made this decision because (1) $C_H$ values from Eq. 15 are based on measured steepness, (2) we thoroughly compare our Eq. 15 estimates with contact migration rates measured in our models, and (3) measured contact

migration rates would not be available for a real stream. Although $tolerance$ can be set in such a way that simulations with X$^2$ values under some threshold are defined as acceptable, we do not use X$^2$ in that manner here. Instead, we show the X$^2$ values for all Eq. 14 estimates (using 200 $K$ values spaced logarithmically from $10^{-9}$ to $10^{-4}$ m$^{1-2n\theta}$ a$^{-1}$) and focus on the $K$ with the lowest X$^2$ as the best-fit $K$ (i.e., this would be the best estimate for the stream's erodibility). Varying the $tolerance$ would scale the magnitudes of all X$^2$ values, but it would not alter which $K$ value corresponds with the lowest X$^2$. We compare the best-fit

$K$ values in each simulation with the simulation's weak and strong erodibilities ($K_W$ and $K_S$).

Although we use this approach to search for the $K$ value that produces the best agreement between Eq. 14 and Eq. 15 estimates of $C_H$ (where the Eq. 14 estimates use a range of $K$ and the Eq. 15 estimates use measured $k_{sn}$), we do not perform such a search for the slope exponent $n$ value. In each simulation, we calculate Eq. 14 and 15 estimates of $C_H$ using the $n$ value



enforced in each simulation. Using the correct *n* is crucial for estimating the appropriate magnitude and dimensions of
erodibility, but our intention here is only to show how accurately *K* can be estimated. Using the incorrect *n* to estimate the *K*
used in a simulation would involve comparing erodibilities with different dimensions (if *m/n* remains constant). As we
discussed in Sect. 2.2, comparing *K* values is context dependent. We could compare the fluvial relief values expected for
different *K*, or we could compare slope patch migration rates. Attempting to fully explore such considerations, however, would
negatively impact the focus and brevity of this study. Furthermore, we perform these analyses on our numerical models to
inform our analysis of Tank Wash (Sect. 2.6), and our analysis of Tank Wash includes the consideration of multiple *n* values.

**2.6 Analysis of Tank Wash**

We explore the behavior of these rivers in numerical models to develop a framework for quantifying erodibility from
bedrock river morphology. After presenting our numerical model results, we apply the developed framework to Tank Wash
near Hanksville, UT (Fig. 1). We use Google Earth imagery and the nearby 1:62k geologic map of the San Rafael Desert
Quadrangle (which includes the same units; (Doelling et al., 2015)) to infer the map-view positions of contacts near Tank
Wash. We then infer the contact locations along Tank Wash's longitudinal profile by considering both the inferred map-view
contacts and changes in the stream's steepness (Fig. 1b). Channel profile data are taken from 10-m digital elevation models
provided by the United States Geological Survey. There are no contact dip measurements available in the vicinity of Tank
Wash, but based on regional geology, the contact dips are likely relatively low. We evaluate contact dips of -1° and -5° (dipping
in the upstream direction) because the geologic maps (Doelling et al., 2015) and imagery available in the area suggest that
contact dips are likely relatively low. Furthermore, our results will demonstrate the effect of contact dip on these analyses in a
manner that enables extrapolation (e.g., considering if the dip was -2.5° or -10°). Although this approach is far from ideal, our
intention here is only to demonstrate how one could apply the developed framework to real streams. Any accurate analyses
would require detailed field surveys, and such endeavours could be the focus of future work.

After identifying the potential contacts, we (1) divide Tank Wash's profile into reaches separated by the inferred
contact locations and (2) use the average steepness of each reach to estimate kinematic wave speed ($C_H$) values according to
Eq. 15. These $C_H$ estimates are made twice for each reach: once at the minimum drainage area and once at the maximum. We
then take the average of all $C_H$ values within five drainage area bins spaced logarithmically from the lowest to highest drainage
areas. To explore what erodibilities could yield similar results (given the assumed contact dips evaluated), we compare the
average $C_H$ estimates from our approach (Eq. 15) with a range of predictions from the Darling et al. (2020) portrayal of
kinematic wave speed (Eq. 14). We perform this comparison with the $X^2$ Misfit Function (Eq. 16). We evaluate a large range
of *K* for Tank Wash (200 values spaced logarithmically from $10^{-8}$ to $10^{-2}$ m$^{1-2n\theta}$ a$^{-1}$, where $\theta = 0.5$). Because Eq. 14 requires an
estimated rock-uplift rate (i.e., base level fall rate), we use the range of incision rates from the cosmogenic dating of fluvial
terraces along the nearby Fremont River (0.3 to 0.85 mm a$^{-1}$; (Repka et al., 1997; Cook et al., 2009); red circle in Fig. 1a).
Importantly, our results will enable us to consider how the estimated erodibility would change with the assumed base level fall
rate (i.e., considering how different would erodibility be if the incision rate was only 0.15 mm a$^{-1}$ instead of 0.3 mm a$^{-1}$). For
this analysis of Tank Wash, we evaluate *n* values of 0.67 and 1.5 and assume that *m/n* = 0.5.



## 3 Results

### 3.1 Scenario 1: two rock types with $\phi = 0°$

In this section, we present the results for scenario one of our numerical models (Table 1). These simulations use two rock types (weak and strong) with contact dips of 0°. We use the results for scenario one to (1) further explain the dynamics of bedrock river incision through flat-lying strata and (2) test and further explore the semi-analytical framework developed by Perne et al. (2017).

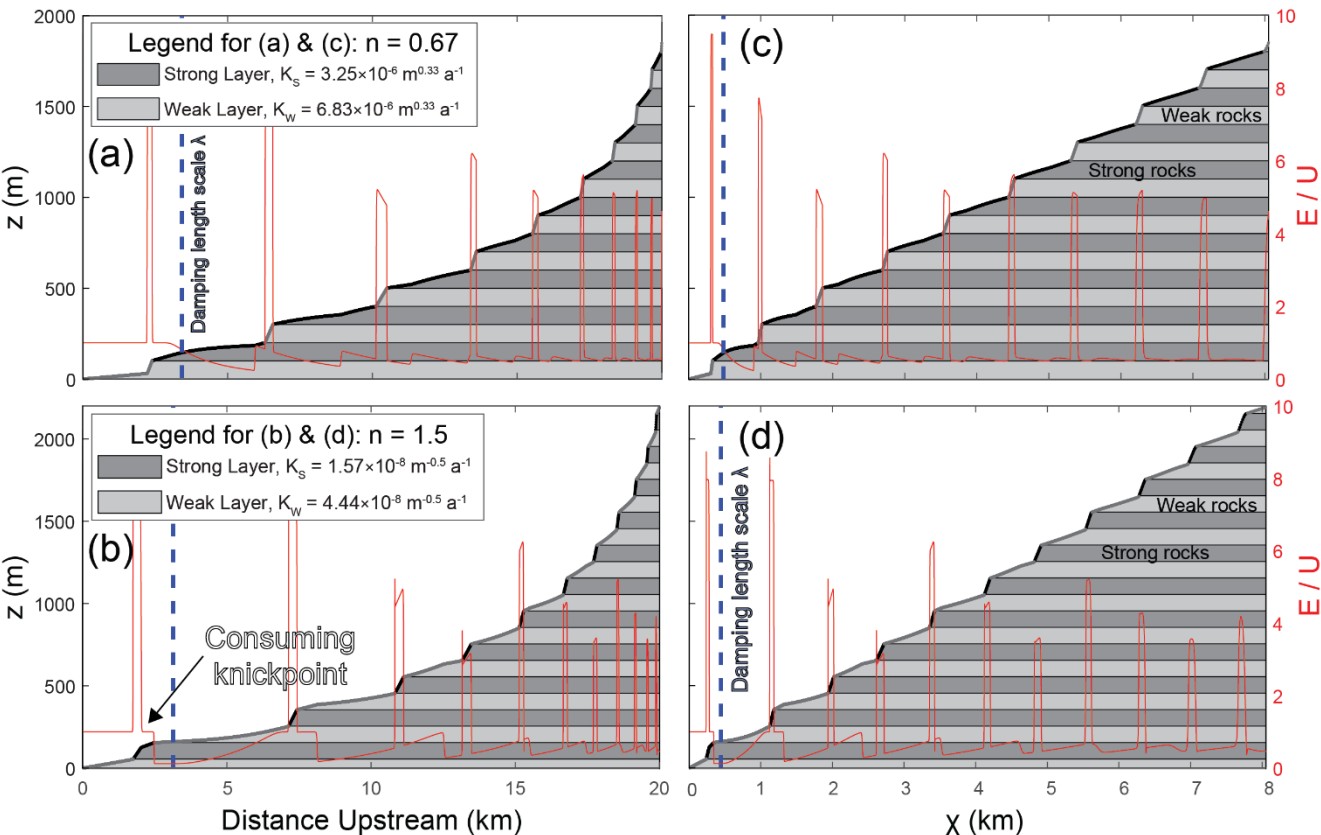

**Figure 3**. Longitudinal profiles (**a-b**) and $\chi$-plots (**c-d**) for two different simulations. The simulation in (**a**) and (**c**) has slope exponent $n$ values of 0.67, while the simulation in (**b**) and (**d**) has $n = 1.5$. Layer thickness $H$ and rock-uplift rate $U$ are 100 m and 0.15 mm a⁻¹ in both simulations, respectively. The position where $\chi$ equals the damping length scale $\lambda$ (Eq. 7) is shown as a dashed blue line. Note the variations in erosion rate $E$.

      Figure 3 shows long profiles (a-b) and $\chi$-plots (c-d) for two simulations of bedrock rivers underlain by alternating

weak and strong rock layers. The simulation in Figs. 3a and 3c has $n = 0.67$ and $K^*$ of ~9.5 (weak layer's erodibility $K$ is ~2.1 times higher than strong layer's $K$), while the simulation in Figs. 3b and 3d has $n = 1.5$ and a $K^*$ of ~0.13 (weak layer's $K$ is ~2.8 times higher than the strong layer's $K$). Both simulations use a rock-uplift rate ($U$) of 0.15 mm a⁻¹ and layer thicknesses ($H$) of 100 m. Note that the erosion rates normalized by rock-uplift rate ($E / U$) are shown as red lines. We show the location where $\chi = \lambda$ (damping length scale; Eq. 7) as a dashed blue line. Like the streams near Hanksville, UT and those simulated by





Perne et al. (2017), these streams have a stepped appearance. As we discussed in Sect. 2.4, when $n < 1$ (Figs. 3a and 3c) reaches

underlain by the weaker rocks have higher channel slopes and erosion rates. When $n > 1$ (Figs. 3b and 3d), reaches underlain

by the stronger rocks have higher channel slopes and erosion rates. Note that we highlight a consuming knickpoint near the

outlet in Fig. 3b, which is relevant to the damping length scale ($\lambda$) discussed in Sect. 2.3. This consuming knickpoint will reach

the top of the reach within the strong unit right before $\chi = \lambda$ (dashed blue line), demonstrating the purpose of the damping

length scale. To explain why the erosion rate variations in Fig. 3 occur, we now examine the contact migration rates within

each of these simulations.

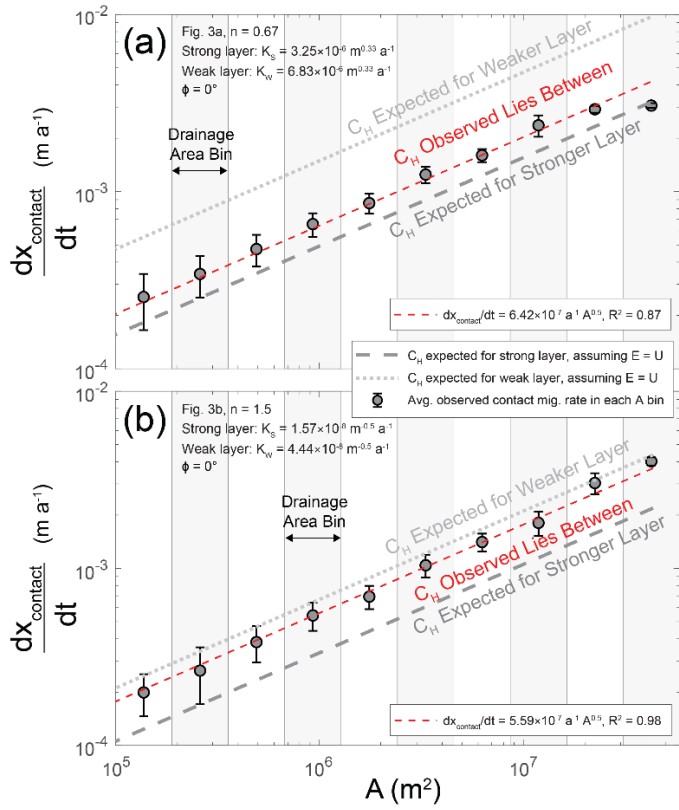

**Figure 4.** (**a**) Contact migration rates ($dx_{contact}/dt$) vs drainage area ($A$) for the simulation shown in Figs. 3a and 3c. (**b**) Contact migration rates vs drainage area for the simulation shown in Figs. 3b and 3d. The slope exponent ($n$) and strong and weak erodibilities ($K_s$ and $K_W$) are shown in the upper left of each subplot. Note that contact dip ($\phi$) here is 0°. Each subplot has dotted and dashed lines for the kinematic wave speeds estimated for the weak and strong layers, respectively, if the erosion rate ($E$) was equal to rock-uplift rate ($U$). The erosion rates in each layer do not conform to this assumption, however, and instead vary so that kinematic wave speed is maintained at a moderate value between the dashed and dotted lines.

Figures 4a and 4b show contact migration rates ($dx_{contact}/dt$) versus drainage area ($A$) for the simulations in Figs. 3a

and 3b, respectively. We show the average measured contact migration rates (gray circles with black outlines) within ten

drainage area bins; the vertical bars for each circle represent the standard deviation of $dx_{contact}/dt$ within the corresponding

drainage area bin. In Figs. 4a and 4b, the light gray dotted line represents the kinematic wave speed (Eq. 8) expected if only

the weak layer was present and all erosion rates were equal to the rock-uplift rate. Similarly, the dark gray dashed line shows



the kinematic wave speed expected if only the strong layer was present and all erosion rates were equal to the rock-uplift rate.

If only one rock type was present, one could think of these wave speeds as the upstream migration rate of bedding planes within the units, as rock uplift carries the units up the stream profile. The measured contact migration rate lies somewhere between these two endmembers. This finding is highlighted by the dashed red lines, which are a power-law functions fit between the observed contact migration rates and the drainage areas at the center of each bin (assuming a drainage area exponent of $m/n$, which is 0.5 here). While the fact that contact migration rates fall between the two extremes shown in each

subplot may sound straightforward, the reason for this result is not intuitive.

These dynamics occur because channel slopes on both sides of the contact interact to drive the system towards equal retreat rates and kinematic wave speeds ($C_H$) across the contact. For example, consider a contact with a weak unit situated beneath a strong unit. The stream segment in the weak unit may initially erode at a higher rate, undercutting the strong unit and forcing the contact further upstream. Importantly, the response of the strong unit depends on slope exponent $n$ in the stream

power model. When $n > 1$, higher erosion rates in the weak unit will cause a consuming knickpoint (Royden and Perron, 2013) to migrate into the strong unit situated above. The strong unit responds rapidly in this case, keeping pace with the weak unit by eroding at a higher rate. This response is so effective that the contact's migration leads to a reduction in channel slope within the weak unit (i.e., lengthening each reach within the weak unit), decreasing the weak unit's erosion rate. When $n < 1$, however, there is no consuming knickpoint. Instead, the initially higher erosion rate in the weak unit causes an erosional signal

that migrates more slowly through the strong unit above. A stretch zone (Royden and Perron, 2013) initially forms at the base of the strong unit (i.e., a convex-upwards knickzone). Instead of rapidly adjusting to keep pace with the weak unit, in this case the strong unit slows down the contact's migration. The stretch zone in the strong unit is then replaced by a reach of low steepness. This transition occurs because the combination of undercutting by the weak unit and resistance from the strong unit leads (1) to higher channel slopes and erosion rates within the weak unit and (2) lower channel slopes and erosion rates within

the strong unit (i.e., due to lengthening of each reach within the strong unit).

Although we discussed these dynamics in qualitative terms above, we will now we discuss them with a stronger focus on contact migration rates (Fig. 4) and kinematic wave speed ($C_H$). To maintain equal retreat rates, reaches within the weaker layer develop a lower $C_H$ value, relative to what would be expected if they were eroding at the rock-uplift rate (dotted gray lines in Fig. 4). Conversely, reaches within the stronger layer develop a higher $C_H$ value. When $n < 1$, Eq. 8 shows that $C_H$ is

inversely proportional to erosion rate $E$. Because of this relationship, reaches in the weak layer achieve a lower $C_H$ (i.e., slow down) by increasing $E$ when $n < 1$. Similarly, reaches in the strong unit achieve a higher $C_H$ (i.e., speed up) by decreasing $E$ when $n < 1$. Due to such behaviors, when $n < 1$ the stream has higher steepness and erosion rates within the weak unit and subdued steepness and erosion rates within the strong unit. The opposite is true when $n > 1$; reaches in the weak unit obtain a lower $C_H$ value (i.e., slow down) by decreasing $E$ and reaches in the strong unit obtain a higher $C_H$ value (i.e., speed up) by

increasing $E$.



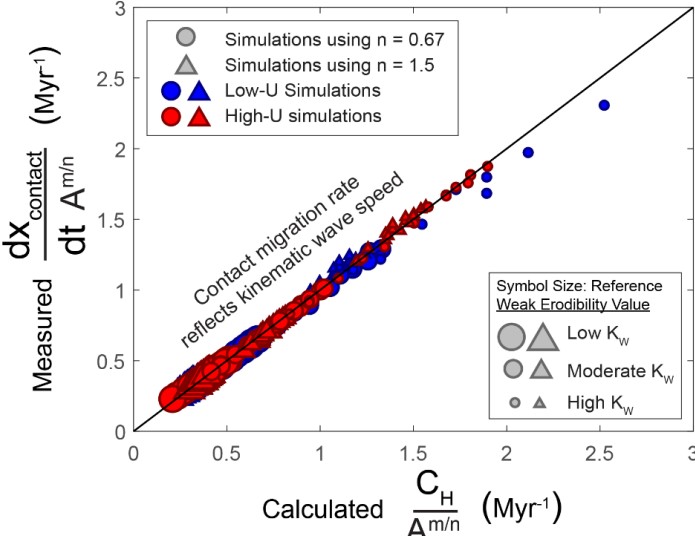

**Figure 5**. Estimated kinematic wave speeds ($C_H$) and measured contact migration rates ($dx_{contact}$ / $dt$) for all simulations in scenario one (Table 1). The low-$U$ scenarios have $U = 0.15$ mm a$^{-1}$ while the high-$U$ scenarios have $U = 0.3$ mm a$^{-1}$. Symbol size represents the reference weak erodibility used ($K_W$; Table 1), with large symbols representing low $K_W$ and small symbols representing high $K_W$. The x-axis values are calculated using Eqs. 8 and 12.

To assess if the theory developed by Perne et al. (2017) is applicable across the parameter space explored in scenario one, we compared kinematic wave speeds calculated with Eqs. 8 and 12 with contact migration rates ($dx_{contact}/dt$) measured in our models (Fig. 5). Note that in Fig. 5, both metrics have been normalized by drainage area raised to the $m/n$; as shown in Fig. 4, contact migration rates change with drainage area. Because Fig. 5 shows all simulations in scenario one, layer thickness for the weak layer ($H_W$) and strong layer ($H_S$) vary from (1) 100 m for both layers, (2) 150 m for both layers, (3) $H_W = 75$ m and $H_S = 150$ m, and (4) $H_W = 150$ m and $H_S = 75$ m. These $H$ values are used in all combinations of reference erodibility for the weak layer ($K_W$, represented by symbol size) and rock-uplift rate ($U$, represented by color). For each weak layer erodibility, a wide range of strong layer erodibilities are also assessed (Sect. 2.2). Despite all of these changes, the kinematic wave speeds predicted using the framework from Perne et al. (2017) (Eqs. 8 and 12) serve as excellent portrayals of the contact migration rates in our numerical models. These findings indicate that when contacts are horizontal, contact migration rates reflect the kinematic wave speeds of the surrounding stream reaches. Furthermore, erosion rate variations like those in Fig. 3 occur so that kinematic wave speeds are equal on either side of a contact, allowing kinematic wave speed to consistently increase with drainage area as shown in Fig. 4.

**3.2 Scenario 2: three rock types with $\phi = 0°$**

In this section, we examine the results for scenario two (Table 1). Like scenario one, scenario two only considers horizontal contacts (contact dip $\phi = 0°$). Unlike scenario one, however, scenario two utilizes three rock types (weak, medium, and strong). Our intention is to test if the equations presented by Perne et al. (2017) still hold when there are more than two rock types, because real streams usually incise through strata that are far more complicated than those considered in scenario one.




Earth **Surface** Dynamics Open Access
Discussions

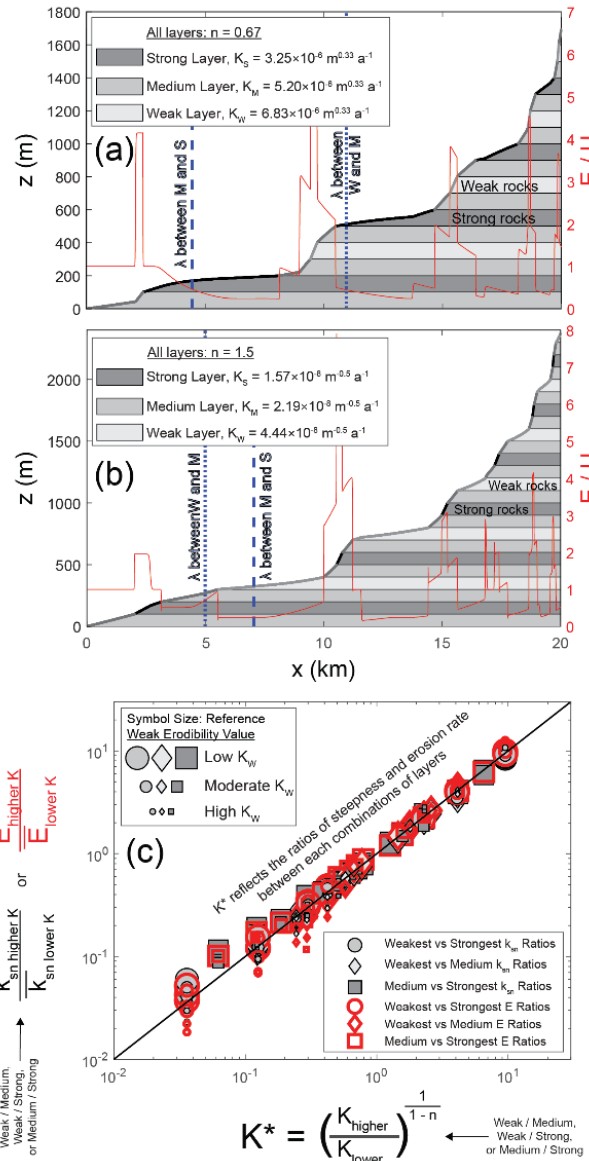

**Figure 6**. Longitudinal profiles (**a-b**) for two simulations using three rock types. The scenario is (**a**) has $n = 0.67$, while the scenario in (**b**) has $n = 1.5$. (**c**) Ratios of steepness $k_{sn}$ and erosion rate $E$ for all simulations in scenario 2 (Table 1). Black and gray symbols represent steepness ratios, while red symbols represent erosion ratios. Layer combinations are represented with symbol shape (e.g., weak layer vs medium layer or weak layer vs strong layer). For the different combinations of the weak, medium, and strong layers, the $k_{sn}$, $E$, and $K$ values corresponding with the more erodible layer are always in the numerator (e.g., $E$ medium / $E$ strong).

Figures 6a and 6b show long profiles with three rock types of equal thickness (100 m) and $n$ values of 0.67 (Fig. 6a) and 1.5 (Fig. 6b). Figure 6c shows ratios of the average steepness values ($k_{sn}$) and erosion rates ($E$) within different rock types (e.g., $k_{sn}$ of weak layer / $k_{sn}$ of strong layer) for all simulations in scenario two. Note that Fig. 6c shows $K^*$ and the ratios of average $k_{sn}$ and $E$ for the three rock types used, so we designate the layer combination (weak vs strong, weak vs medium,



medium vs strong) by symbol shape. For example, the red diamonds represent $E$ in the weak layer / $E$ in the medium layer. In all cases, the $k_{sn}$, $E$, and $K$ values for the more erodible of the two layers is in the numerator of the ratio (e.g., for the ratio of $k_{sn}$ weak / $k_{sn}$ medium, $K^*$ is calculated using $K_W$ and $K_M$). Because the steepness and erosion ratios in Fig. 6c follow a 1:1 relationship with $K^*$, these results are consistent with Eqs. 9c and 10.

560         These results suggest the theory developed by Perne et al. (2017) for bedrock river incision through horizontal strata still applies when there are more than two rock types. When there is an additional rock type, the channel slopes and erosion rates within the additional rock type will adjust to allow for a consistent trend in kinematic wave speed across the profile. Here, the medium layer is the additional rock type, relative to the simulations in scenario one. For example, Fig. S6 shows the contact migration rates for the simulations in Figs. 6a and 6b; despite differing erodibilities, contact migration rates and $C_H$ increase

as a power-law function of drainage area. The fact that steepness ratios and erosion-rate ratios are both well represented by $K^*$ (Fig. 6c) follows from Eqs. 9c and 10, which were derived by setting the kinematic wave speeds within two rock types equal to each other.

### 3.3 Scenarios 3 and 4

### 3.3.1 General morphologic results of nonzero contact dips

570         Before we test the framework developed by Darling et al. (2020) for bedrock river incision through nonhorizontal strata, we discuss the general morphologic implications of nonzero contact dips. River morphology can be significantly altered by even slight changes in contact dip (Figs. 7 and 8). Figure 7 shows the long profiles (a-b) and $\chi$-plots (c-d) for two simulations with $n = 0.67$ and the same erodibility values ($K$) used in Fig. 3a, but with slight dips to the contacts. One simulation has contacts dipping upstream at 2.5° ($\phi = -2.5°$; Figs. 7a and 7c), and the other simulation has contacts dipping downstream at

2.5° ($\phi = 2.5°$; Figs. 7b and 7d). Although the strong and weak erodibilities in Fig. 7 are the same as those in Fig. 3a, the morphologies of these streams are quite distinct. For example, although the simulations use the same erodibilities, rock-uplift rates, layer thicknesses, and drainage areas, the maximum river elevations in Figs. 3a, 7a, and 7b are about 1800 m, 2300 m, and 1700 m, respectively. Indeed, such pronounced changes in river erosion and morphology for deviations in contact dip of only 2.5° away from horizontal bedding planes highlight the importance of contact dip in river morphology. Note that in the

$\chi$-plots in Fig. 7, the apparent contact dip in $\chi$-space ($\phi_\chi$; Eq. 5) varies along the profile. When contacts dip upstream ($\phi < 0°$) $\phi_\chi$ is negative, and when contacts dip downstream ($\phi > 0°$) $\phi_\chi$ is positive. In both $\chi$-plots, however, the absolute value of $\phi_\chi$ approaches zero with increasing $\chi$ (i.e., the contacts seem to bend and almost become horizontal).

        The two simulations in Fig. 8 have $n = 1.5$ and the same erodibility ($K$) values used in Fig. 3b, but with nonzero contact dips. Figures 8a and 8b are the long profiles for these simulations, while Figs. 8c and 8d are the $\chi$-plots. Because we

examined simulations with the same absolute contact dips in Fig. 7 ($|\phi| = 2.5°$), we now show simulations with dissimilar contact dips: Figs. 8a and 8c has contacts dipping 10° upstream ($\phi = -10°$), while the simulation in Figs. 8b and 8d have contacts dipping 1° downstream ($\phi = 1°$). These two simulations with $n = 1.5$ (Fig. 8) also have distinct morphologies relative to similar simulations with a contact dip of 0° (Fig. 3b). For example, the maximum elevations for Figs. 3b, 8a, and 8b are about 2200 m, 2600 m, and 2000 m, respectively, even though the erodibilities used are the same. These distinctions highlight the role of

Earth **Surface**
**Dynamics**
Discussions

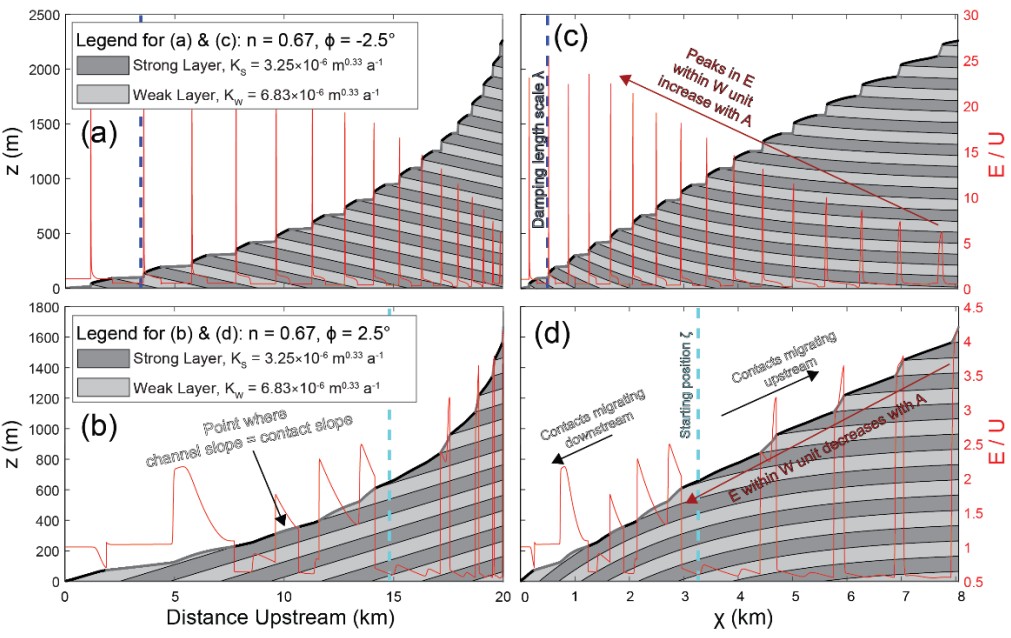

**Figure 7.** Longitudinal profiles (**a-b**) and χ-plots (**c-d**) for two different simulations with $n = 0.67$. The simulation in (**a**) and (**c**) has contact dip $\phi = -2.5º$, while the simulation in (**b**) and (**d**) has $\phi = 2.5º$. Layer thickness $H$ and rock-uplift rate $U$ are 100 m and 0.15 mm a$^{-1}$ in both simulations, respectively.

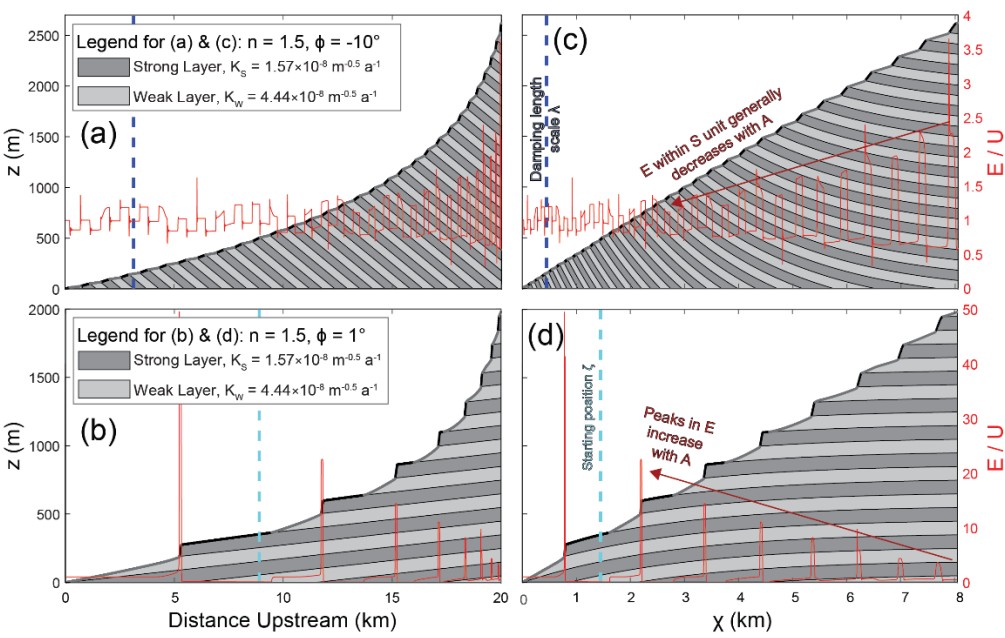

**Figure 8.** Longitudinal profiles (**a-b**) and χ-plots (**c-d**) for two different simulations with $n = 1.5$. The simulation in (**a**) and (**c**) has contact dip $\phi = -10º$, while the simulation in (**b**) and (**d**) has $\phi = 1º$. Layer thickness $H$ and rock-uplift rate $U$ are 100 m and 0.15 mm a$^{-1}$ in both simulations, respectively.





contact dip in bedrock river morphology. Even a contact dip of 1° causes a striking departure from the behaviors expected for horizontal bedding (Figs. 8b and 8d). For example, even though the simulation in Fig. 8b is the same as that in Fig. 3b except

for a contact dip of 1°, these two simulations have very different spatial patterns in erosion and steepness. With $n > 1$ and contacts dipping 1° in the downstream direction (Fig. 8b), erosion rates are no longer relatively uniform within each rock type. Instead, there are sharp peaks in erosion rate near contacts (i.e., consuming knickpoints), and these peaks increase in magnitude with distance downstream.

### 3.3.2 Contact migration rates for nonzero contact dips

We now examine contact migration rates in simulations with nonzero contact dips (scenarios 3 and 4; Table 1). Scenario 3 has contacts dipping in the upstream direction ($\phi < 0°$), while scenario 4 has contacts dipping in the downstream direction ($\phi > 0°$). In this section, we utilize these simulations' contact migration rates to explain the bedrock river dynamics discussed in Sect. 3.3.1 above (i.e., why erosion rates change along the profile).

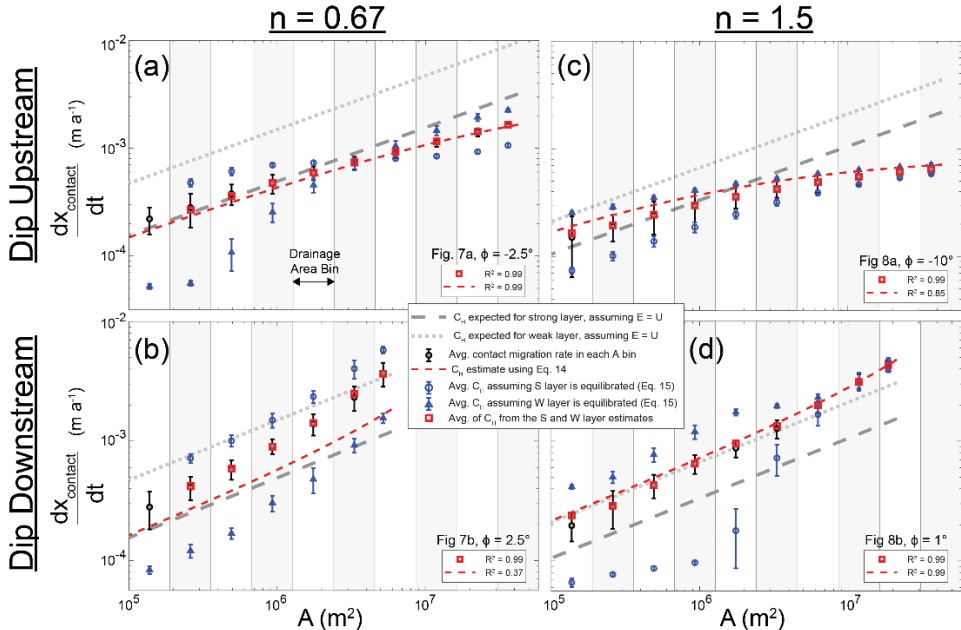

**Figure 9**. Contact migration rates ($dx_{contact}/dt$) vs drainage area ($A$) for the four simulations shown in Figs. 7 and 8. All simulations here have layer thickness $H = 100$ m. The simulations have slope exponent ($n$) and weak and strong erodibility values ($K_W$ and $K_S$) of (**a-b**) $n = 0.67$, $K_W = 6.83 \times 10^{-6}$ m$^{0.33}$ a$^{-1}$, and $K_S = 3.25 \times 10^{-6}$ m$^{0.33}$ a$^{-1}$ and (**c-d**) $n = 1.5$, $K_W = 4.44 \times 10^{-8}$ m$^{-0.5}$ a$^{-1}$, and $K_S = 1.57 \times 10^{-8}$ m$^{-0.5}$ a$^{-1}$. The simulations have contact dip ($\phi$) values of (**a**) $\phi = -2.5°$, (**b**) $\phi = 2.5°$, (**c**) $\phi = -10°$, and (**d**) $\phi = 1°$. Note that these Eq. 15 estimates of kinematic wave speed ($C_H$) use all $k_{sn}$ measured over the 10 Myr duration

for each simulation.

Figure 9 shows contact migration rate ($dx_{contact}/dt$) versus drainage area for each of the four simulations in Figs. 7 and 8. Like in Fig. 4, the measured contact migration rates in Fig. 9 are gray circles with black outlines and vertical bars representing the standard deviation of $dx_{contact}/dt$ within each drainage area bin. The red dashed line, red symbols, and blue symbols are the kinematic wave speed estimates made with Eqs. 14 and 15; we will address these estimates after exploring general trends in





the observed $dx_{contact}/dt$ data. The simulations in Figs. 9a and 9c have contacts dipping in the upstream direction (Figs. 7a and 7b), while the simulations in Figs. 9c and 9d have contacts dipping in the downstream direction (Figs. 8a and 8b). Unlike simulations with a flat-lying stratigraphy, these simulations with nonzero contact dips have contact migration rates that do not follow a consistent power-law relationship with drainage area. Because our results in Sect. 3.1 established that contact migration rates are a reflection of kinematic wave speeds, these results show that for nonzero contact dips, the rate at which

kinematic wave speed ($C_H$) increases with drainage area is a function of contact dip. These results confirm the implications of Eq. 14 (Darling et al., 2020); contacts dipping in the upstream direction (Figs. 9a and 9c) decrease the rate at which kinematic wave speed increases with drainage area, and contacts dipping in the downstream direction (Figs. 9b and 9d) increase the rate at which kinematic wave speed increases with drainage area.

We now discuss the drivers of these relationships. Even when contact dip is nonzero, a stream can adjust to maintain

equal retreat rates on either side of a contact in the same manner discussed in Sect. 3.1 (i.e., an effective adjustment within the strong layer when $n > 1$ vs resistance in the strong layer when $n < 1$). When contact dip is nonzero, however, the important distinction is that the contact's geometry relative to the channel can limit or enhance the contact's migration in the upstream direction. The migration rates of contacts that dip in the downstream direction ($\phi > 0°$) will be enhanced because the erosion of the contact will cause the contact to be exposed further upstream. Conversely, the migration rates of contacts that dip in the

upstream direction ($\phi < 0°$) will be limited because the contact plane recedes deeper into the subsurface in the upstream direction. The magnitudes of these influences depend on the contrast between channel slope and contact slope, and this contrast changes with drainage area.

Now, we explain these dynamics by examining the relationship between kinematic wave speed and erosion rate when contact dips are nonzero. Even when contact dip is nonzero, slope exponent $n$ controls the relationship between a stream's

kinematic wave speed ($C_H$) and erosion rate ($E$; Eq. 8). We will first discuss cases with $n < 1$. When $n < 1$, then $C_H$ is inversely proportional to $E$ (Eq. 8). If contacts are dipping upstream ($\phi < 0°$) so that the growth rate of $C_H$ with drainage area must gradually decrease (e.g., Fig. 9a), then a stream with $n < 1$ will achieve this decreased growth in $C_H$ by having large spikes in channel slope and erosion rate near the contacts (Fig. 7c). The magnitudes of the erosion rate spikes increase with distance downstream because the growth rate of $C_H$ will decrease with distance downstream (to maintain equal retreat rates on either

side of a contact). These spikes in erosion will also occur in the weak unit near the contact, as the weaker unit's larger erodibility requires a greater reduction in kinematic wave speed. Conversely, if contacts are dipping downstream ($\phi > 0°$) then the growth rate of $C_H$ with drainage area will increase (e.g., Fig. 9b). A stream with $n < 1$ will achieve this acceleration in the growth of $C_H$ with drainage area through a reduction in the weak unit's erosion rate with drainage area (Fig. 7d). We have highlighted such erosion rate variations in Fig. 7.

If slope exponent $n > 1$, the same trends in kinematic wave speed will occur across the profile (i.e., changing growth rate with drainage area). Importantly, however, the same $C_H$ requirements will be met with erosion rate variations that are the opposite of those occurring when $n < 1$. This distinction lies in the fact that when $n > 1$, $C_H$ is proportional to $E$ (Eq. 8). For





example, if contacts dip upstream ($\phi < 0°$) and $n > 1$ then a decrease in the growth rate of $C_H$ with drainage area (Fig. 9c) is achieved through a decrease in erosion rate with drainage area (Fig. 8c). If contacts instead dip in the downstream direction ($\phi > 0°$) and $n > 1$, then an increase in the growth rate of $C_H$ with drainage area (Fig. 9d) is achieved through an increase in erosion rates near contacts (Fig. 8d; the undercutting of the strong unit by the weak unit creates a consuming knickpoint, and the positive contact dip allows the knickpoint to migrate farther upstream). We have highlighted such erosion rate variations in Fig. 8. These spatial variations in erosion rate occur so the stream maintains equal retreat rates on either side of each contact. When contact dips are nonzero, the magnitude of contact slope relative to channel slope changes as a function of drainage area. This relationship causes the influence of contact migration on channel slope to change with drainage area.

Overall, the spatial patterns in kinematic wave speed (increasing or decreasing growth rate with drainage area) depend on contact dip and are independent of slope exponent $n$, but $n$ controls the spatial patterns in erosion rate that accomplish the required patterns in kinematic wave speed. Although contact migration rates continue to reflect the pattern of kinematic wave speeds across the profile when dip is nonzero, the erosion rates that are representative of these kinematic wave speeds can be highly localized near the contacts (e.g., consuming knickpoints in Fig. 8d).

### 3.3.3 Kinematic wave speeds for nonzero contact dips

In this section, we test both (1) the framework for kinematic wave speed ($C_H$) developed by Darling et al. (2020) for bedrock river incision through nonhorizontal strata (Eq. 14) and (2) our approach for estimating kinematic wave speed with measured steepness values (Eq. 15). We test these approaches by comparing contact migration rates measured in our models with $C_H$ estimates made with Eqs. 14 and 15 for all simulations in scenarios 3 and 4 (Table 1).

The red dashed lines in Fig. 9 are $C_H$ estimates made with Eq. 14. These curves capture how the growth rate of kinematic wave speed with drainage area must accelerate or decelerate for positive and negative contact dips, respectively. While capturing the overall trends, the approach of Darling et al. (2020) (Eq. 14) does deviate in some situations. For example, the $R^2$ values for these estimates (relative to the average contact migration rates in each drainage area bin) are high for upstream dipping contacts and $n < 1$ (Fig. 9a) but quite low for downstream dipping contacts and $n < 1$ (Fig. 9b). Such deviations occur because Eq. 14 was derived on the assumption that the less steep layer has erosion rates equal to the rock-uplift rate. This assumption is why, for example, the red dashed line starts out along the dark gray dashed line at low drainage areas in Fig. 9b. Like in Fig. 4, this gray dashed line represents the kinematic wave speeds expected for the strong layer if its erosion rate was equal to the rock-uplift rate. In this simulation, however, the strong layer's erosion rates at low drainage areas are lower than the rock-uplift rate (Fig. 7d). This deviation from the assumptions made by Darling et al. (2020) causes the modelled contact migration rates to be higher than the red dashed line (Fig. 9b).

The blue circles and triangles in Fig. 9 are kinematic wave speed ($C_H$) estimates made using Eq. 15, which utilizes the steepness ($k_{sn}$) measured within each drainage area bin. The blue circles are the average Eq. 15 estimates made with the strong layer's $k_{sn}$, while the blue triangles are the average Eq. 15 estimates made with the weak layer's $k_{sn}$. Vertical bars on each value represent the standard deviation of Eq. 15 estimates within each drainage area bin (due to variations in $k_{sn}$; these examples use the entire 10 Myr of recorded $k_{sn}$, rather than only the final time step). When $n < 1$ (Figs. 9a and 9b), the $C_H$





estimates made with the strong layer are higher than measured contact migration rates. Conversely, when $n > 1$ (Figs. 9c and 9d) the $C_H$ estimates made with the weak layer are higher than measured contact migration rates. These estimates are too high because the corresponding layer tends to have an erosion rate $E$ lower than the rock-uplift rate $U$ (strong layer when $n < 1$, 690 weak layer when $n > 1$). The other layer's erosion rate will be higher than $U$, however, and taking the average of Eq. 15 estimates made for the strong and weak layers produces an extremely accurate depiction of the measured contact migration rates. For example, the red squares in Fig. 9 are the average of the weak and strong layers' Eq. 15 estimates of kinematic wave speed. Relative to the measured contact migration rates, these red squares have $R^2$ values of 0.99 in each subplot of Fig. 9. Note that there is no Eq. 15 estimate in the lowest drainage area bin of Fig. 9b because, in this case ($n < 1$), reaches in the weak 695 layer tend to be less steep. Because the reaches in the weak unit are less steep, they span a longer horizontal distance along the channel. As a result, an entire reach within the weak would not fit within the lowest drainage area bin in this simulation.

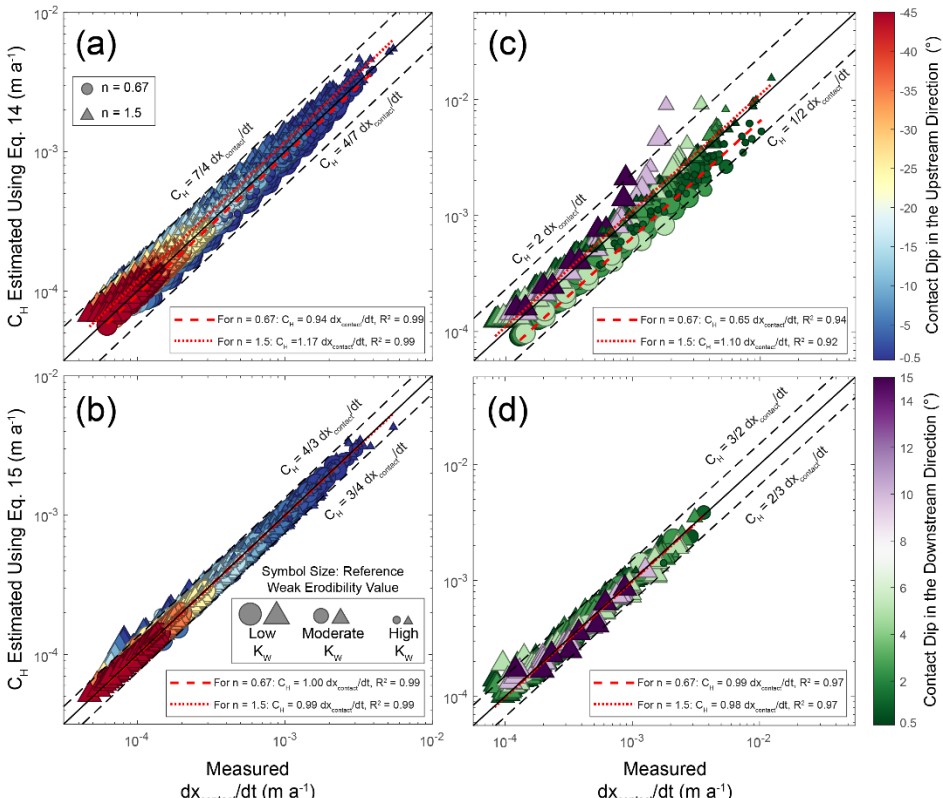

**Figure 10**. Contact migration rates measured in our models ($dx_{contact}/dt$) vs. kinematic wave speeds ($C_H$) estimated using Eqs. 14-15. Subplots (**a-b**) show results for scenario 3 (contacts dipping upstream, $\phi < 0°$) and subplots (**c-d**) show results for 700 scenario 4 (contacts dipping downstream, $\phi > 0°$). Subplots (**a**) and (**c**) use Eq. 14, while subplots (**b**) and (**d**) use Eq. 15. Red dashed and dotted lines are linear regressions for results with $n$ values of 0.67 and 1.5, respectively. Dashed lines show the minimum and maximum values for most values in each subplot, with labels denoting the corresponding relationships between contact migration rate and kinematic wave speed.

Figure 10 shows estimates of kinematic wave speed ($C_H$) made with Eqs. 14-15 relative to contact migration rates 705 ($dx_{contact}/dt$) measured in all simulations with nonzero contact dips (scenarios 3 and 4; Table 1). Each point represents the





measured $dx_{contact}/dt$ and estimated $C_H$ within one drainage area bin (e.g., Fig. 9). Circles represent simulations with $n = 0.67$,
while triangles represent simulations with $n = 1.5$. Symbol size represents the erodibility for the weak layer ($K_W$), with large
points used for low $K_W$ and small points used for high $K_W$. Subplots (a-b) show results for contacts dipping upstream ($\phi < 0°$),
while subplots (c-d) show results for contacts dipping downstream ($\phi > 0°$). Each subplot has linear regressions between
measured contact migration rates and estimated kinematic wave speeds for simulations with $n = 0.67$ and $n = 1.5$ (dashed and
dotted red lines, respectively). Each subplot also has a solid black line representing a 1:1 relationship between measured contact
migration rates and estimated $C_H$. Furthermore, dashed black lines denote the general maximum and minimum values in each
subplot (e.g., $C_H = 1/2\ dx_{contact}/dt$ and $C_H = 2\ dx_{contact}/dt$). Note that these Eq. 15 estimates only use the $k_{sn}$ recorded in one
model timestep, rather than the entire 10 Myr of recorded values. This choice was motivated by the data limitations for real
streams. Figure S7 shows Eq. 15 estimates using all recorded $k_{sn}$; the accuracy is similarly high, except there are more data
points.

Overall, both Eq. 14 and Eq. 15 provide highly accurate portrayals of contact migration rates (Fig. 10). Even though
Eq. 14 can be less accurate at times (Fig. 9), the equation's performance is consistently good across the parameter space
explored in scenarios 3 and 4 (Fig. 10). We show how Eqs. 14 and 15 can be combined to estimate erodibility in Sect. 3.3.5,
but we will first more thoroughly explore how erosion rates vary for nonzero contact dips (Sect. 3.3.4).

**3.3.4 Erosion rate variations for nonzero contact dips**

In this section, we explore the variations in erosion rate that occur for nonzero contact dips in greater detail. Through
these analyses, we develop three-dimensional regressions between $K^*$, $\ln(|\phi_\chi|)$, and $E_W / U$ (Eq. 13). Recall that $\phi_\chi$ is the contact
dip in $\chi$-space (nondimensional; change in contact elevation / change in the overlying river's $\chi$ values) and $K^*$ is a term
describing erodibility contrasts (Eq. 9c). The purpose of this analysis is to (1) highlight the magnitude of erosion variations
that occur when contact dip is nonzero and (2) relate these variations to both drainage area and contact dip.

Figure 11 shows the erosion rates in the weak layer ($E_W$) normalized by rock-uplift rates ($U$) for all simulations with
$n = 1.5$ and contacts dipping upstream ($\phi < 0°$; scenario 3). There are gray shadows on the $\ln(|\phi_\chi|)$-$K^*$ plane situated directly
beneath each point. Each $E_W / U$ and $\ln(|\phi_\chi|)$ value is the average within one drainage area bin (e.g., Fig. 9) in one simulation.
$K^*$ represents the contrast between the weak and strong layers' erodibilities in that simulation (($K_W / K_S$)$^{1/(1-n)}$; Eq. 9c). Note
that when $n > 1$, lower $K^*$ values indicate larger erodibility contrasts. Here, the simulation's contact dip is represented by color,
while the magnitude of the weak layer's erodibility ($K_W$; Table 1) is represented by symbol size (large symbol for low $K_W$,
small symbols for high $K_W$). Note that there are no small symbols for the highest $K^*$ because those simulations always had
damping length scales $\lambda$ that were too large (Sect. 2.3). Also note that there are greater deviations from the overall trend in
simulations using the highest $K_W$ (small symbols).

Across the parameter space used for the simulations in Fig. 11, there is a consistent trend in $E_W / U$. At large erodibility
contrasts (for $n > 1$, low $K^*$), the weak layer erodes at a rate much lower than the rock-uplift rate. At small erodibility contrasts
(for $n > 1$, high $K^*$), the variations in erosion rate are more subdued. The magnitude of $\ln(|\phi_\chi|)$ also influences erosion rate. For
example, when $\ln(|\phi_\chi|)$ is high all $E_W / U$ values approach one. At low $\ln(|\phi_\chi|)$ values, the erosion rates approach those expected



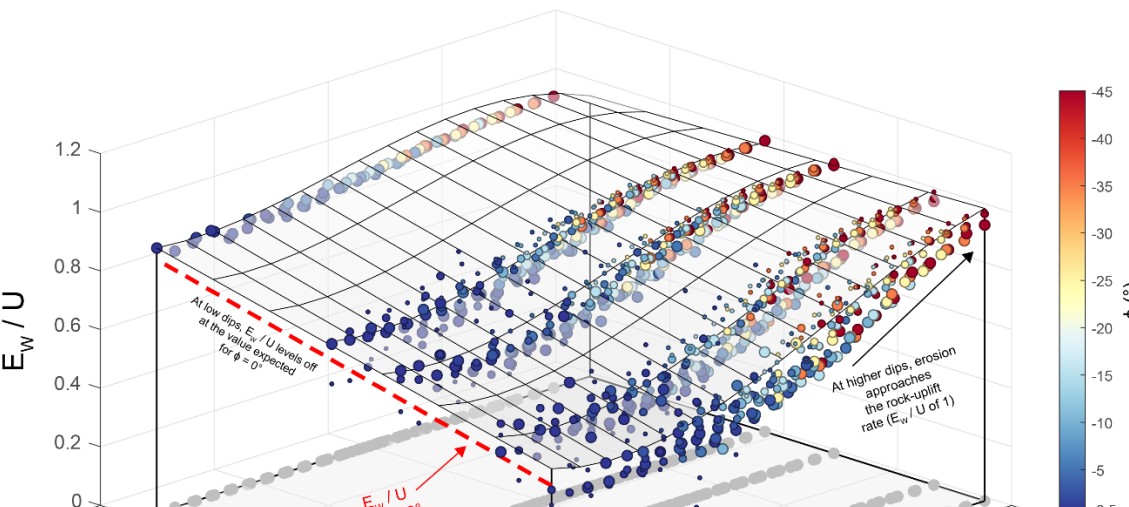


**Figure 11**. Variations in the average erosion rate in the weak layer ($E_W$) normalized by rock-uplift rate ($U$) with both the logarithm of the absolute contact dip in $\chi$-space ($\ln(|\phi_\chi|)$) and the enforced $K^*$ (Eq. 9c) for simulations with $n = 1.5$ and contacts dipping upstream ($\phi < 0°$). Note that symbol size represents the reference weak erodibility ($K_W$), with smaller points corresponding with higher $K_W$ values. Also note that the $E_W / U$ and $\ln(|\phi_\chi|)$ values here are the mean values taken within

logarithmically spaced drainage area bins (e.g., Fig. 9). A regression is fit to all data ($R^2 = 0.81$): $E_W / U = (-2.2 \times 10^{-3} \ln(|\phi_\chi|)^3)$ $+ (-1.7 \times 10^{-2} \ln(|\phi_\chi|)^2 K^*) + (-4.6 \times 10^{-3} \ln(|\phi_\chi|)^2) + (-1.4 \times 10^{-1} \ln(|\phi_\chi|) K^*) + (9.7 \times 10^{-2} \ln(|\phi_\chi|)) + (2.6 \times 10^{-1} K^*) + 8.5 \times 10^{-1}$. Note that points are colored by $\phi$ and have shadows directly beneath them. The red dashed line represents the erosion rates expected if the contact dip was 0° (Eqs. 8 and 12).

for a contact dip of 0° (dashed red line; Eqs. 8 and 12). Note that $\ln(|\phi_\chi|)$ increases with distance downstream along a profile

(i.e., at higher drainage areas, the channel slope is smaller relative to contact slope). Because of this relationship, $\ln(|\phi_\chi|)$

represents a combination of (1) position along the stream profile and (2) the magnitude of contact dip. Higher contact dips

and/or larger drainage areas increase $\ln(|\phi_\chi|)$, causing $E_W / U$ to approach values of one. Lower contact dips and/or smaller

drainage areas decrease $\ln(|\phi_\chi|)$, causing $E_W / U$ to approach the values expected for a contact dip of 0° (red dashed line). The

multilinear regression shown in Fig. 11 ($R^2 = 0.81$) captures these relationships between erosion rate ($E_W / U$), erodibility

contrasts ($K^*$), and both drainage area and contact dip ($\ln(|\phi_\chi|)$). The residuals for this regression are shown in Fig. S8.

We present Fig. 11 to demonstrate general trends in erosion rate for nonzero contact dips, but we provide similar

figures for other combinations of unit type (i.e., strong or weak), slope exponent $n$, and contact dip in Figs. S9-S19. We always

fit multilinear regression between $K^*$, $\ln(|\phi_\chi|)$, and the weak layer's normalized erosion rate ($E_W / U$), but we also fit a multilinear

regression for the strong unit ($E_S / U$) when $n = 0.67$ and contacts dip downstream ($\phi > 0°$). Otherwise, the erosion rates within





the strong unit are not captured well by such regressions (e.g., greater deviations from the overall trend). The other regressions
we evaluate vary in their accuracy, with $R^2$ values ranging from 0.32 (Fig. S11) to 0.92 (Fig. S13). Like the regression in Fig.
11, however, these regressions always highlight the roles of $K^*$ and $\ln(|\phi_\chi|)$ in setting the magnitudes of erosion rate variations.
$K^*$ reflects what the erosion rates would be if the contact dip was 0° (e.g., red dashed line in Fig. 11), while variations in $\ln(|\phi_\chi|)$
values control how different portions of the profile approach the erosion rates expected for 0° (reflecting the combined
influence of drainage area and contact dip).

There is an important distinction to note regarding one of the other regressions, however. When contacts dip
downstream ($\phi > 0°$) and $n = 1.5$, there is a change in how erosion rates vary with $\ln(|\phi_\chi|)$ (Fig. S11). Because the growth rate
of kinematic wave speed increases with drainage area when $\phi > 0°$ (e.g., Fig. 9d), consuming knickpoints naturally form when
$n > 1$ (e.g., Fig. 8b). These consuming knickpoints cause erosion rates to dramatically increase with $\ln(|\phi_\chi|)$ (e.g., $E_W > 10\ U$ at
high $\ln(|\phi_\chi|)$ and high erodibility contrasts). This dramatic increase in erosion rate with $\ln(|\phi_\chi|)$ only occurs for those conditions
($n > 1$ and $\phi > 0°$) due to the formation of consuming knickpoints.

To summarize the results for this section, when contact dips are nonzero the erosion rates within each unit change as
a function of contact dip, drainage area (which both set $\phi_\chi$), and erodibility contrasts ($K^*$). Assuming that the erosion rate is
equal to the base level fall rate (Eq. 14) will cause one to overestimate kinematic wave speed ($C_H$) when $n > 1$ (e.g., Fig. 9c)
and underestimate $C_H$ when $n < 1$ (e.g., Fig. 9b). Note that our intention is not to assign theoretical significance to these three-
dimensional regressions, but only to use them to highlight how contact dip, drainage area, and erodibility contrasts influence
the erosion rates for nonzero contact dips.

### 3.3.5 Estimating erodibility in our numerical models using nonzero contact dips

In this section, we evaluate how accurately the erodibility ($K$) in our numerical models can be estimated using channel
steepness ($k_{sn}$) without apriori information on erodibility. We demonstrated that Eqs. 14 and 15 are generally accurate (Fig.
10), but now we demonstrate how the two equations can be combined to estimate erodibility. The purpose of this analysis is
to provide context for our analysis of Tank Wash near Hanksville, UT (Sect. 3.4). We will attempt to quantify erodibility using
the stream steepness observed along Tank Wash, so our intention here is to test how accurately one can estimate erodibility in
numerical models where the true erodibility values are known.

Figures 12 and 13 summarize how the $K$ values we use in our numerical models can be estimated by combining Eqs.
14 and 15. Figure 12 focuses on weak erodibilities ($K_W$) used in our numerical models, while Fig. 13 focuses on the strong
erodibilities ($K_S$). Figures 12a, 12b, 13a, and 13b show the $X^2$ values calculated between (1) the $C_H$ estimates made with Eq.
14 for a range of erodibilities (200 values spaced logarithmically from $10^{-9}$ to $10^{-4}$ $m^{1-2n\theta}$ $a^{-1}$, where $\theta = 0.5$) and (2) the $C_H$
estimates made with Eq. 15 for the $k_{sn}$ recorded in each simulation's final timestep. We chose to use only one timestep of
790    recorded $k_{sn}$ values due to the data limitations for real rivers, but Figs. S20 and S21 are versions of Figs. 12 and 13 that use all
$k_{sn}$ recorded over the 10 Myr for each simulation. The x-axes of Figs. 12a, 12b, 13a, and 13b represent the entire range of $K$
values assessed ($10^{-9}$ to $10^{-4}$ $m^{1-2n\theta}$ $a^{-1}$) normalized by the actual weak or strong erodibility used in that simulation ($K_W$ in Fig.



Earth **Surface**
**Dynamics**
Discussions



12, $K_S$ in Fig. 13). Each line represents one simulation (blue for simulations with $\phi < 0°$, red for simulations with $\phi > 0°$). Each of these lines has a minimum X$^2$ value, and invariably this minimum occurs near an Assessed $K$ / Enforced $K$ value of one.



**Figure 12**. Comparison of best-fit $K$ values in our numerical models to the weak erodibility ($K_W$) used in each simulation. (**a-b**) X$^2$ Misfit Function values for kinematic wave speeds ($C_H$) estimated using Eq. 14, the enforced contact dip ($\phi$), and a wide range of $K$ values (200 points spaced logarithmically from $10^{-9}$ to $10^{-4}$ m$^{1-2n\theta}$a$^{-1}$, where $\theta = 0.5$) relative to the Eq. 15 estimates of $C_H$. (**c-d**) Comparison between the best-fit $K$ and the $K_W$ enforced in the simulations. Subplots (**a**) and (**c**) show results for $n = 0.67$, while subplots (**b**) and (**d**) show results for $n = 1.5$.





**Figure 13**. Comparison of best-fit $K$ values in our numerical models to the strong erodibility ($K_S$) used in each simulation. (**a-b**) $X^2$ Misfit Function values for kinematic wave speeds ($C_H$) estimated using Eq. 14, the enforced contact dip ($\phi$), and a wide range of $K$ values (200 points spaced logarithmically from $10^{-9}$ to $10^{-4}$ m$^{1-2n\theta}$a$^{-1}$, where $\theta = 0.5$) relative to the Eq. 15 estimates of $C_H$. (**c-d**) Comparison between the best-fit $K$ and the $K_S$ enforced in the simulations. Subplots (**a**) and (**c**) show results for $n = 0.67$, while subplots (**b**) and (**d**) show results for $n = 1.5$.

Figures 12c, 12d, 13c, and 13d are histograms showing the distributions of these minimum $X^2$ values relative to the weak and strong erodibilities ($K_W$ in Fig. 12, $K_S$ in Fig. 13). The x-axes for these histograms (best-fit $K$ / enforced $K$) use bin sizes of 0.1. The best-fit $K$ is almost always between $K_W$ and $K_S$. For example, the best-fit $K$ / enforced $K_W$ values in Fig. 12 are generally less than one (some exceptions in Fig. 12d). Conversely, the best-fit $K$ / enforced $K_S$ values in Fig. 13 are always greater than one. The best-fit $K$ is only slightly higher than $K_S$ when $n < 1$ (Fig. 13c), but it can be much higher than $K_S$ when $n > 1$ (e.g., best-fit $K > 3$ $K_S$; Fig. 13d). This larger contrast between the best-fit $K$ and $K_S$ occurs for $n > 1$ because (1) the strong layer tends to be steeper when $n > 1$ and (2) the strong layer has larger deviations in erosion rate relative to the rock-





uplift rate when $n > 1$ (Figs. S15 and S17). In contrast, the weak layer has higher steepness and erosion rates when $n < 1$ (Figs.

3, 7, S9, and S13). Although the best-fit $K$ is generally close to $K_W$ when $n < 1$ (e.g., typically within 50 %; Fig. 12c), it is more

consistently close to $K_S$ (e.g., typically within 30 %; Fig. 13c). These results suggest that an erodibility estimated using this

approach is more likely to be representative of the rock type with lower steepness. We can estimate the erodibility of the less

steep layer (weak layer when $n > 1$, strong layer when $n < 1$) because that layer tends to have smaller deviations in erosion rate

relative to the steep layer (Figs. 11 and S8-S19). These results highlight the potential to estimate erodibility using observed

channel steepness ($k_{sn}$), known contact dips ($\phi$), and a known base level fall rate ($U$).

**3.8 Analysis of Tank Wash**

In this section, we apply the methods developed in this study (Sect. 3.3.5) to Tank Wash (Fig. 14). Figure 14a shows

inferred contacts labeled with purple lines. The units clearly dip to the west/southwest here, which is generally in the upstream

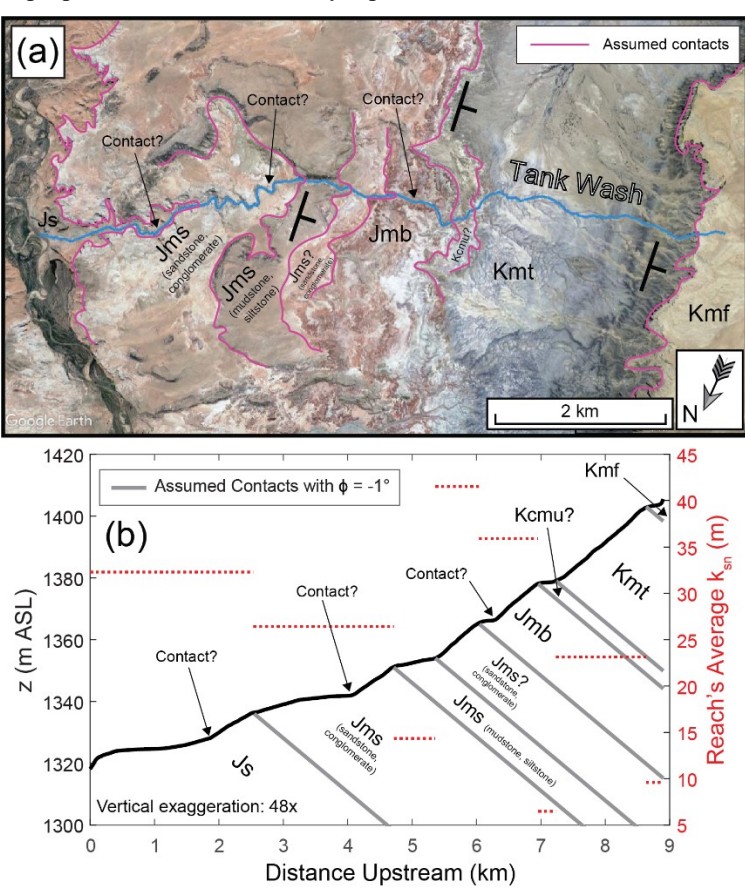

**Figure 14**. Overview of Tank Wash, a stream near Hanksville, UT. Subplot (**a**) is a Google Earth image (© Google Earth)
with the stream and geology shown. We show potential contacts locations as purple lines. Subplot (**b**) is a longitudinal profile
for Tank Wash with the assumed contacts from (**a**) shown as grey lines. We assume a contact dip ($\phi$) of -1° for these contacts.
The average steepness ($k_{sn}$) for each reach situated between these contacts is shown in red. The lithologies are as follows: **Kmf**
is brown sandstone and mudstone; **Kmt** is grey shale, siltstone, and mudstone; **Kcmu** and **Jmb** are color-banded siltstone,
claystone, mudstone, and shale; **Jms** contains both red-brown mudstone and siltsone as well as light-yellow-gray lenticular
sandstone and conglomerate; and **Js** is red-brown siltstone, sandstone, and gypsum.





direction for Tank Wash. Using the (1) potential contacts identified using Google Earth and a nearby geologic map (Doelling et al., 2015) and (2) changes in steepness along Tank Wash, we then estimated the contact locations along the stream's longitudinal profile (Fig. 14b). Tank Wash flows across a wide range of lithologies, including sandstone, mudstone, shale, and

siltstone, and these units likely offer different levels of resistance to fluvial erosion. In Fig. 14b, these contacts are projected into the subsurface using an assumed contact dip ($\phi$) of -1° (note the vertical exaggeration). Each of the potential contacts we identified occur at a change in channel steepness, but there are three locations where steepness changes but we could not confidently infer a change in lithology. We highlight those three locations in Figs. 14a and 14b. Those steepness changes could indicate, for example, changes in rock strength within one unit. It is important to note, however, that our models suggest

changes in steepness can occur both at contacts and some distance away from contacts (e.g., Figs. 7a and 8b).

Figure 15a is a slope-area plot, with the inferred contact locations shown as vertical dashed lines and the average steepness values of the reaches between each contact shown as dotted lines. We used these average steepness values to estimate kinematic wave speeds ($C_H$) using Eq. 15 (Figs. 15b and 15c). Note that the results in Fig. 15b use an assumed contact dip of -1°, while those in Fig. 15c use an assumed contact dip of -5°. Also note that Figs. 15b and 15c have two y-axes: one for $C_H$

estimates with base level fall rate $U = 0.3$ mm a$^{-1}$ (blue y-axis on right) and another with $U = 0.85$ mm a$^{-1}$ (red y-axis on left). These base level fall rates are based on the cosmogenic dating of nearby fluvial terraces (Repka et al., 1997; Cook et al., 2009). The patterns in the data are the same, with $C_H$ magnitudes being scaled by the assumed $U$. We estimated two kinematic wave speeds for each stream reach defined by the inferred contacts: one $C_H$ at the lowest drainage area of the reach, and one at the highest. In Figs. 15b and 15c, each pair of $C_H$ estimates is connected by a dashed line. If Tank Wash has achieved a dynamic

equilibrium (i.e., variations in steepness and erosion rate are due to contact migration rather than changes in base level fall), our results have shown that the true $C_H$ values should lie between the highest and lowest estimates (Eq. 15). To pursue this balance, we therefore evaluated the average $C_H$ within five bins spaced logarithmically from the lowest to the highest drainage areas. These average $C_H$ values are shown in Figs. 15b and 15c as black squares with horizontal bars extending across the corresponding drainage area bin. Although this approach is not the same as taking the average of Eq. 15 estimates made

separately for weak and strong units (e.g., Fig. 9), our intention is only to pursue moderate values situated between the highest and lowest $C_H$ estimates in Figs. 15b and 15c.

The dashed and dotted black lines in Figs. 15b and 15c are the Eq. 14 kinematic wave speeds for the best-fit erodibility values. The dashed lines use a slope exponent $n$ of 0.67, while the dotted lines use an $n$ of 1.5. These best-fit $K$ values are those with the lowest X² Misfit Function Value (Eq. 16) relative to the average Eq. 15 estimates of $C_H$ (black squares in Figs. 15b

and 15c). The X² values for all $K$ are shown in Fig. 16. The $K$ values range from $10^{-8}$ to $10^{-2}$ m$^{1-2n\theta}$ a$^{-1}$ (where $\theta = 0.5$), and for each combination of contact dip, base level fall rate, and $n$, there is one $K$ corresponding with the minimum X². Varying the contact dip does not change the selection of a best-fit $K$ because contact dip scales both the Eq. 14 and Eq. 15 estimates (Figs. 14b and 14c). Varying either base level fall rate or slope exponent $n$ does, however, alter best-fit $K$ values (Fig. 16). Although the two different base level fall rates ($U$) we assessed produce distinct best-fit $K$ values, these rates cover quite a large range

in $U$ (they differ by almost a factor of 3). Furthermore, the uncertainty involved in erodibility can be orders of magnitude



Earth **Surface**
**Dynamics**
Discussions



(Stock and Montgomery, 1999), so being able to gain some quantitative constraints on $K$ for a reasonable range of erosion rates is certainly an advancement. Based on results from our numerical models (Figs. 12-13), we would argue that these erodibilities are (1) likely between the highest and lowest erodibilities present and (2) likely more representative of the unit with the lowest steepness. Because Tank Wash has its lowest steepness values in areas we infer to have mudstone and/or

siltstone (Kmf, Kmcu, and Jms; Fig. 14), it is possible these erodibilities are more representative of those units. Because one

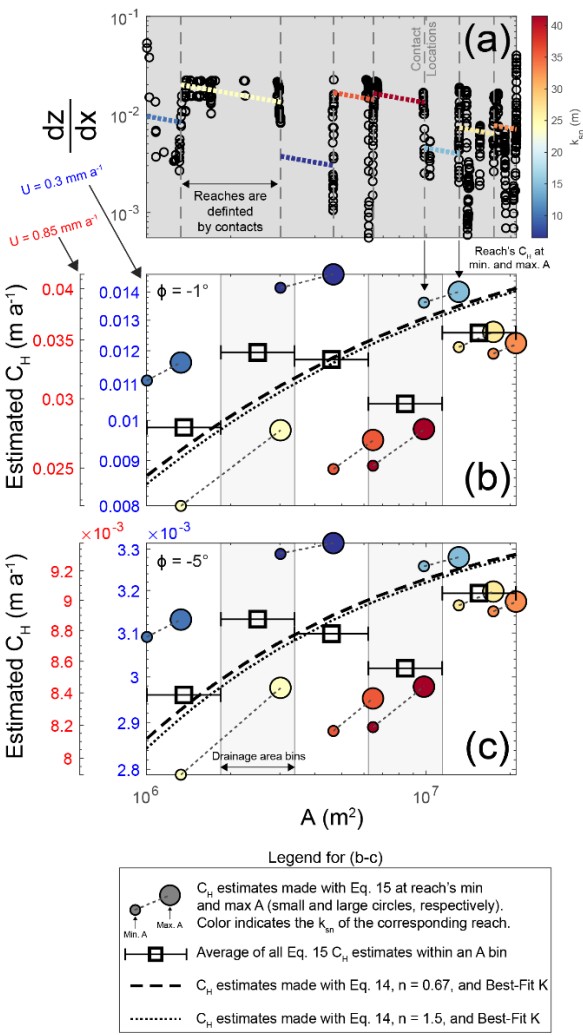

**Figure 15**. Preliminary analysis for Tank Wash. (**a**) Slope-area plot with gray dashed lines for each presumptive contact identified in Fig. 14. The average steepness ($k_{sn}$) of each reach between contacts is shown as colored dotted lines. (**b-c**) Kinematic wave speed values ($C_H$) estimated with Eq. 15 using the average $k_{sn}$ values, rock-uplift rate $U = 0.3$ mm a$^{-1}$ or $U =$

0.85 mm a$^{-1}$ (separate axes), and contact dip ($\phi$) values of either (**b**) -1° or (**c**) -5°. In addition to the $C_H$ estimates made using river morphology (Eq. 15), two sets of dashed lines are shown in (**b-c**); these curves use the approach of Darling et al. (2020) (Eq. 14), and represent the best-fit erodibility ($K$) values (Fig. 16) for each combination of slope exponent $n$, $\phi$, and $U$. The best-fit $K$ values are as follows: (1) for $n = 0.67$, $U = 0.3$ mm a$^{-1}$, and both $\phi$ values, $K = 4.45×10^{-5}$ m$^{0.33}$ a$^{-1}$; (2) for $n = 0.67$, $U = 0.85$ mm a$^{-1}$, and both $\phi$ values, $K = 1.26×10^{-5}$ m$^{0.33}$ a$^{-1}$; (3) for $n = 1.5$, $U = 0.3$ mm a$^{-1}$, and both $\phi$ values, $K = 3.92×10^{-6}$

m$^{-0.5}$ a$^{-1}$; and (4) for $n = 1.5$, $U = 0.85$ mm a$^{-1}$, and both $\phi$ values, $K = 1.11×10^{-5}$ m$^{-0.5}$ a$^{-1}$.



Earth **Surface**
**Dynamics**
Discussions



may intuitively expect mudstone and siltstone to be weaker than other units present, like sandstone, then the lower steepness within those seemingly weaker units could suggest a slope exponent *n* that is greater than one. Although there are considerable uncertainties involved in this inference, Darling et al. (2020) also found that streams incising through sedimentary units in southern Utah had morphologies that were consistent with *n* > 1.

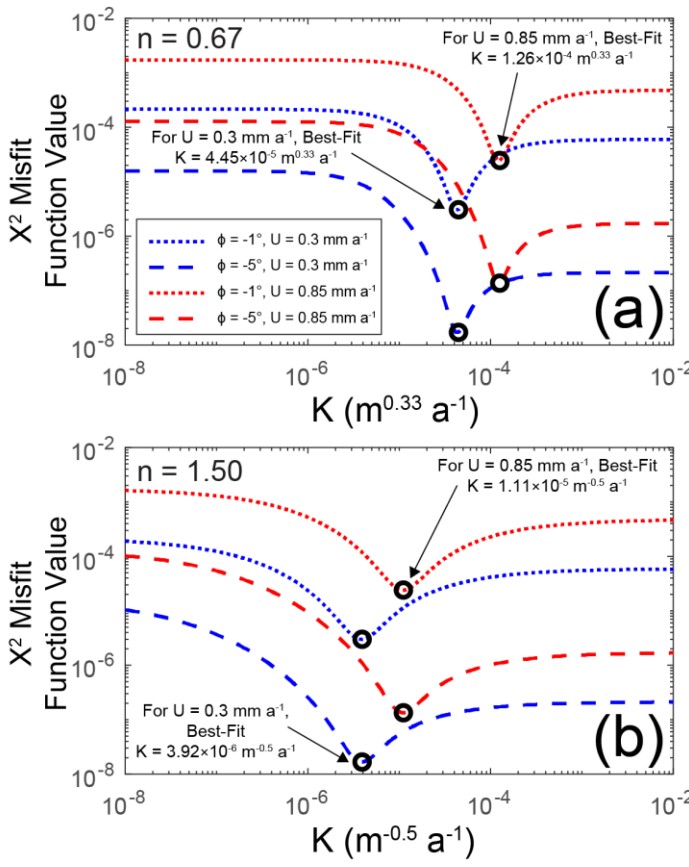

**Figure 16**. $X^2$ Misfit Function values for kinematic wave speed ($C_H$) estimates made with Eq. 14 and a wide range of erodibility (*K*) values relative to the $C_H$ estimated for Tank Wash with Eq. 15 (Fig. 15). Subplot (**a**) shows results for *n* = 0.67, while subplot (**b**) shows results for *n* = 1.5. Note that the best-fit erodibility *K* (situated at the lowest $X^2$) does not change with assumed contact dip ($\phi$) here, although it does change with the assumed slope exponent *n* and rock-uplift rate (*U*). Also note that the $X^2$ values are lower for $\phi$ = -5° because a greater dip in the upstream direction causes a smaller range of kinematic wave speeds (e.g., Fig. 9c). The misfit for this reduced range can be smaller, but these lower $X^2$ values do not indicate that the dip is closer to -5° than to -1°.

**4 Discussion**

We have shown here that for rivers incising into layered rocks, channel steepness, erosion rates, and contact migration rates continue to reflect rock strength differences across a wide range of contact dips. Changes in river erosion and morphology with dip are predictable, enabling the perturbation of contact migration to be exploited for insight rather than avoided or ignored. This finding also emphasizes that the effects of mixed lithologies must always be weighed when considering (1)



bedrock river morphology and (2) erosion rate estimates based on techniques like detrital cosmogenic nuclide analysis (Darling et al., 2020).

**4.1: Evaluation of previous work**

Our findings demonstrate that when contacts are horizontal ($\phi = 0°$), the equations developed by Perne et al. (2017) (Eqs. 9c and 12) remain applicable even if there are large changes in parameters like erodibility, rock-uplift rates, and layer thicknesses. We have also shown here that when there are more than two rock types, contact dips of zero, and the stream has achieved a dynamic equilibrium, stream reaches in each rock type will erode at rates that provide a uniform trend in kinematic

wave speed ($C_H$) across the profile. Whether a unit erodes above or below rock-uplift rate $U$ depends on the slope exponent $n$ value and the magnitude of the layer's erodibility in relation to the other layers' erodibilities. We have also shown that the erosion rates expected for a contact dip of 0° are important when contact dips are nonzero; portions of the profile with high channel slope (relative to the absolute value of contact slope) can approach these erosion rates even when contact dip is nonzero (i.e., low $\ln(|\phi_\chi|)$ values in Fig. 11).

Our results also demonstrate that the equation for kinematic wave speed (Eq. 14) developed by Darling et al. (2020) is a robust depiction of contact migration rates for nonzero contact dips (Fig. 10). Although this approach can be less accurate at times (Fig. 9), the magnitudes of such deviations are small in relation to the potential variations in erodibility. As a result, we were able to use their approach to accurately estimate the erodibilities used in our numerical models (Figs. 12-13). These authors also developed an expression for the erosion rate required for reaches in a strong unit to have the same kinematic wave

speed as the weak unit (which was assumed to erode at the base level fall rate). Although we focus on how erosion rates in the weak unit can deviate from the base level fall rate (Fig. 11), the expression developed by Darling et al. (2020) could be applied with modified erosion rates in the weak unit.

**4.2: Influence of rock strength contrasts on the pursuit of equilibrium**

Because our results demonstrate the complexity of patterns in channel steepness and erosion rate along rivers eroding

through layered rocks, these complexities must be considered within the adjustment of landscapes. If contact migration perturbs channel slopes and erosion rates in real landscapes, then these perturbations could significantly impact a landscape's adjustment to changes in tectonics or climate. These impacts could shorten or lengthen timescales of transient adjustment. For example, this study and previous work (Darling et al., 2020) have shown that contacts dipping upstream ($\phi < 0°$) can cause the growth rate of kinematic wave speed with drainage area to decrease (e.g., Figs. 9a and 9c). This decrease in the growth of $C_H$

with drainage area might increase the response times of a river, perhaps causing the river to effectively have a lower erodibility at higher drainage areas. Conversely, when contacts dip downstream there is an acceleration in the growth rate of $C_H$ with drainage area (e.g., Figs. 9b and 9d). This acceleration in the growth of $C_H$ may cause the river to effectively have a higher erodibility at higher drainage areas. As demonstrated by these examples, erodibility could effectively be a function of both rock type and drainage area ($K = f(x,A)$), even if the weak and strong erodibilities do not change with drainage area. Otherwise,

the combined influence of drainage area and contact dip on erosion rate (Fig. 11) could make higher and lower elevations of a landscape respond differently to changes in climate. For example, erosion rates may deviate farther from base level fall rates





at low drainage areas where channel slope is high relative to the absolute value of contact slope (i.e., low $\ln(|\phi_\chi|)$), and such deviations would cause these drainage areas to respond differently to changes in precipitation. Such dynamics remain to be demonstrated, but the fact remains that geomorphologists' expectations regarding landscape evolution are shaped by the idea

that erosion rates will be controlled by base level fall and climate; our findings and previous work (Forte et al., 2016; Perne et al., 2017; Darling et al., 2020) suggest that rock strength contrasts can also influence erosion rates.

The role of rock strength contrasts in a landscape's adjustment to climate and tectonics is important, but changes in rock strength contrasts may also be capable of driving landscape transience (Forte et al., 2016). We focus here on a stratigraphy with a repeating pattern of rock types, and despite spatial variations in erosion rate, the streams eventually achieve a dynamic

equilibrium so that the range of elevations is constant with time. The stratigraphic record, however, is far more complicated than the stratigraphy we use. For example, if a stream had equilibrated to incision through two rock types but then a third rock type was exposed, the subsequent changes in erosion rate could begin a long-lasting transient (Forte et al., 2016). Alternatively, landscape transience could be caused by changes in unit thickness, even if the rock types remain the same (i.e., a change towards thicker weak units and thinner strong units would alter fluvial relief). Because we demonstrate that even slight changes

in contact dip can cause marked changes in river behavior and morphology (Figs. 3, 7, and 8), one might also imagine that changes in contact dip with time due to tectonic folding could cause temporal changes in the influence of rock strength contrasts (in addition to the base level changes due to tectonic activity). Furthermore, because we demonstrate that drainage area can influence erosion rates along rivers incising through layered rocks (e.g., $\ln(|\phi_\chi|)$ values in Fig. 11 are a proxy for drainage area), this consideration could be important for drainage reorganization (Willett et al., 2014). Indeed, the presence of layered rocks

can exert a strong influence on drainage network evolution (Ward, 2019; Sheehan and Ward, 2020, 2020b), and spatial variations in erosion rate due to rock strength contrasts would further complicate both drainage divide migration and stream capture. To summarize these considerations, the presence of rock strength contrasts might make dynamic equilibrium more of a moving target. With factors like climate and tectonics changing over different timescales, the additional consideration of contrasts in rock strength could make dynamic equilibrium a more elusive consequence for landscape evolution.

**4.3: Exploring the role of rock strength contrasts in other models of fluvial erosion**

The motivation for this study was to understand the implications of the stream power model for variations in channel slope, erosion rate, and kinematic wave speed for different combinations of contact dip and erodibility contrasts. Before we can test if the common form of the stream power model accurately depicts rivers like those near Hanksville UT, we must understand the morphological implications of the stream power model. We find here that, according to the common form of

the stream power model, variations in channel slope due to rock strength contrasts are set by: (1) the rock-uplift rate ($U$); (2) the contact dip $\phi$; (3) the spatial distribution of drainage area; (4) the magnitude of each layer's erodibility ($K$), as this distribution controls the moderate kinematic wave speeds that the rivers will settle upon (Fig. 4); and (5) the contrast among these erodibilities as represented by $K^*$. The last two points may seem redundant, but the distinction lies in how two river systems may have the same $K^*$ value, but the river systems may have erodibilities of differing magnitudes. Overall, our





965 preliminary analysis of Tank Wash suggests the stream does conform to predictions from the stream power model (e.g., the kinematic wave speeds estimated with Eq. 15 can be approximated using a best-fit erodibility and Eq. 14).

It is important to note, however, that the stream power model (Whipple and Tucker, 1999) depicts shear stress or unit stream-power with simplifying assumptions regarding: (1) the variations in discharge and channel width with drainage area; (2) the role of sediment cover (Sklar and Dietrich, 2004); (3) the presence of erosion thresholds (DiBiase and Whipple, 2011; 970 Lague, 2014); and (4) the use of a single geomorphically representative flow. Clearly, the stream power model is not fully correct, but that failing does not imply the model cannot be useful. Gasparini and Brandon (2011) did show, for example, that the saltation-abrasion model (Sklar and Dietrich, 2004), a generalized abrasion model (Parker, 2004), and a transport-limited model could all be expressed in a form consistent with the stream power model. If the results of different portrayals of fluvial erosion can be sufficiently expressed as power-law relationships involving drainage area and channel slope, then our findings 975 may be pertinent to alternative portrayals of fluvial erosion. Clearly, however, there is more work to be done involving the use of other fluvial models. Below, we focus on two considerations we consider to be significant for such modelling efforts: the role of sediment cover and the potential influence of dynamic channel width adjustment.

In this study, we have focused exclusively on detachment-limited rives. Real landscapes will not be purely detachment limited, of course, so the influence of sediment cover on bedrock rivers incising through layered rocks remains an important 980 consideration. For example, Johnson et al. (2009) showed that channel slopes can reflect the characteristics of sediment load rather than bedrock properties. There is likely a threshold in sediment cover over which the dynamics examined here disappear entirely (i.e., the feedbacks between contact migration, channel slope, and erosion rate are hindered by sediment deposition). There could be lower levels of sediment cover, however, where the dynamics we study here still occur albeit at lower magnitudes. Both (1) the extent to which sediment cover must be limited for rock strength contrasts to perturb erosion rates 985 and (2) whether such conditions are likely to occur in real landscapes remain as outstanding questions. Limited sediment cover within all rock types may allow for erosion rate variations, but sediment cover could also covary with rock type. For example, a strong rock type may have limited sediment cover while a weak rock type has more persistent sediment cover. At present, it is unclear whether these feedbacks can occur if one lithology has transport-limited reaches while another lithology has detachment-limited reaches. Future work should explore the erosion rate variations that occur in numerical models that can 990 freely transition between detachment-limited and transport-limited fluvial processes (Davy and Lague, 2009; Shobe et al., 2017; Yanites, 2018).

We use the common form of the stream power model here, and this model is constructed with assumptions regarding the scaling between drainage area and channel width (Whipple and Tucker, 1999). If channel width follows an assumed power-law scaling with drainage area, the only aspect of channel morphology that can be adjusted is channel slope. Previous work, 995 however, shows that dynamic channel width adjustment may play a significant role in the transient adjustment of bedrock rivers (Yanites et al., 2010; Yanites, 2018). If the erosion variations caused by rock strength contrasts could be accommodated by variations in channel width or process efficiency (e.g., abrasion vs plucking; (Hancock et al., 1998)), then detecting the influence of contact migration in longitudinal profiles could be challenging. Although one might suspect that systematic





changes in channel width or erosion processes near contacts should be easily recognizable in the field, the conditions

observable in the field may not always be representative of conditions during geomorphically significant flows (e.g, changes in bedrock exposure and erosion processes during floods; (Hartshorn et al., 2002)). The role of channel width adjustments should be examined in future modelling studies.

**4.4: Application of our approach to real landscapes**

For our simulations, we focus on rivers incising through a stratigraphy with a spatially uniform contact dip. Real

streams can have the apparent dip change over space, however, either through (1) changes in the actual contact dips or (2) changes in the streamflow direction relative to the units' strike. For example, the potential field example shown in Fig. 1 has spatial changes in both contact dip and streamflow directions. Indeed, the stream shown in Fig. 1c has a flow direction that is almost orthogonal to that of Tank Wash (Figs. 1b and 14). We have shown that even slight changes in apparent dip can have a pronounced effect on bedrock river behavior (Figs. 3, 7, and 8), and the different streams near Hanksville may therefore

require separate, unique treatments. Although apparent dip could vary along each stream, these streams are generally incising through the same units. Based on the results from our numerical models, such field examples may therefore represent an opportunity to (1) use different streams to sample a wide range of drainage areas and contact dips and (2) search for the erodibilities that would satisfy these disparate river morphologies (i.e., through a combination of Eqs. 14, 15, and 16). For example, tributary confluences may provide opportunities to corroborate different erodibility estimates for a tributary and the

larger river it flows into.

The results for some of our simulations were not used because the damping length scale $\lambda$ (Eq. 7; (Perne et al., 2017)) occurred at a $\chi$ value that was too high (e.g., beyond the maximum $\chi$; Sect. 2.3). The damping length scale is used to define the distance from base level at which the influence of constant base level forcing disappears (i.e., consuming knickpoints migrating through the strong layer). Upstream of this location, the dynamics of contact migration are fully expressed. One

might wonder if real streams can fail to fully express the influence of contact migration in a similar manner; can the length scales required for these dynamics to develop be too large to occur in a real landscape? Such complications would hinder our ability to extract rock strength information from landscape morphology. One then must wonder, however, where a hard boundary in base level forcing would occur in a real landscape. Our numerical models have a basin outlet that acts as a hard boundary, and the constant base level fall at this boundary is the reason why we use the damping length scale. Perhaps such a

boundary could effectively occur where a bedrock-dominated tributary flows into a larger, transport-limited river. Such a relationship between base level forcing and a tributary's response remains to be demonstrated, however.

There could certainly be complications in applying our method for estimating kinematic wave speed from steepness (Eq. 15) to real streams. For example, such applications would still require independently constrained base-level fall rates or rock-uplift rates ($U$). The patterns in the estimated kinematic wave speeds in Figs. 15b and 15c are the same for different $U$

values, however, so uncertainties in $U$ may not be a limiting factor in these analyses. Although we are able to accurately estimate $C_H$ using the steepness within weak and strong units, $k_{sn}$ values along real streams are notoriously variable (Wobus et al., 2006). Although our results suggest that one might be able to use variations in channel steepness to gain constraints on



slope exponent *n* (i.e., if *n* < 1 or *n* > 1), the pronounced variations in steepness that can occur in real landscapes would likely impede such an effort.

1035       Although our analysis of Tank Wash suggests that such an approach is promising, we acknowledge our analysis of Tank Wash is far from ideal. A rigorous analysis of Tank Wash requires both detailed field surveys of contact dip and more constraints on the spatial patterns of erosion; has the stream achieved a dynamic equilibrium, or is it in a state of transient adjustment due to a change in base level fall rates? Nonetheless, our intention here is to only show how one could take the methods used on our numerical models and apply them to streams in the real world. We have shown that kinematic wave

speeds estimated for a real stream with Eq. 15 can be roughly matched using Eqs. 14 and 16, and that capability could open new research directions for the field of geomorphology. Although the erodibilities estimated using our approach will be between the weakest and strongest erodibilities present (Figs. 12-13), the large uncertainties involved in erodibility make any quantitative constraints quite valuable. Furthermore, one could use this approach to constrain the influence of climate on erodibility (e.g., estimating and comparing erodibilities for two areas with similar lithologies but different climates).

**5 Conclusions**

      We show here that for bedrock rivers incising through layered rocks, rock strength contrasts between different units can alter channel slopes, erosion rates, and kinematic wave speeds along the river profile. We have also shown, however, that the influence of rock strength contrasts is a predictable phenomenon across a range of contact dips and erodibilities. Because rivers set the boundary conditions for hillslopes, the influence of contact migration may extend across entire landscapes

underlain by layered stratigraphy. We show that predictions from previously developed frameworks for streams eroding through horizontal strata (Perne et al., 2017) and for streams eroding through nonhorizontal strata (Darling et al., 2020) are generally robust. Specifically, we show that contact migration rates along bedrock rivers reflect the pattern of kinematic wave speeds along the river profile. When contact dip is 0°, reaches in each unit develop channel slopes and erosion rates that provide a consistent power-law scaling between kinematic wave speed and drainage area. The kinematic wave speeds maintained along

the profile fall between those expected for the weakest and strongest units if the erosion rates were equal to base level fall rates. When contact dip is nonzero, the changing contrast between channel slope and contact slope alters these dynamics entirely. The growth rate of kinematic wave speed with drainage area will increase if contacts dip towards the outlet ($\phi > 0°$) or decrease if contacts dip towards the channel head ($\phi < 0°$). These changes in the growth rate of kinematic wave speed can cause channel slopes and erosion rates in each unit to vary with drainage area. Furthermore, we have developed and tested a

new method for estimating kinematic wave speed from bedrock river morphology (Eq. 15). Importantly, this new approach can be combined with previous work (Darling et al., 2020) to quantify erodibility along bedrock rivers. We demonstrate this approach by applying it to both our numerical models and a stream near Hanksville, UT. While other methods for quantifying erodibility can require a transient response to changes in base level (Larimer et al., 2018), this new method can be applied to streams with steady base level fall. Overall, our findings show that the presence of mixed lithologies must always be considered

when evaluating (1) bedrock river morphology, (2) erosion rate estimates based on detrital cosmogenic nuclide analysis



(Darling et al., 2020), and (3) interpretations of the stratigraphic record (Forte et al., 2016). Finally, this work shows that geomorphologists can advance our field by exploiting bedrock river erosion through layered rocks, even in settings with stable tectonics and climate, in a manner similar to how we would exploit the transient responses of rivers to changes in external forcings (Whipple, 2004).

**Data Availability**

This study's data are available in the IUScholarWorks data repository at Indiana University (http://hdl.handle.net/2022/26085).

**Author contribution**

Both NM and BY designed and conceptualized the research objective of this study. NM was responsible for the methodology, software, and data visualization. BY provided supervision and acquired the funding for this study. Both authors participated
in writing the manuscript.

**Competing interests**

The authors declare that they have no conflict of interest.

**Acknowledgements**

This study was supported by a grant from the National Science Foundation (NSF EAR-1727139). This research was supported
in part by Lily Endowment, Inc., through its support for the Indiana University Pervasive Technology Institute.

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
