# Peer review of "Bedrock River Erosion through Dipping Layered Rocks: Quantifying Erodibility through Kinematic Wave Speed"

_Earth Surface Dynamics, 2021_

## Referee Comment (RC2)

**Review of "Bedrock River Erosion through Dipping Layered Rocks: Quantifying Erodibility through Kinematic Wave Speed" by Mitchell and Yanites (esurf)**

Boris Gailleton - boris.gailleton@gfz-potsdam.de

April 2021

**General comments**

Mitchell and Yanites present a method to explore the impact of uplifting layers of different rock strength on landscape evolution. They use a semi-analytical 1D stream power model to investigate how these differences in rock strength translate to fluvial morphology, the spatial distribution of erosion rates and the $K$ vs $n$ parameters, by expressing the migration of the contacts with a kinematic wave equation. They also apply this method to a real landscape to undertake the difficult exercise of constraining the values of erodibility $K$ for a layered landscape. Overall, the paper nicely outlines how dynamic differential lithology evolving in 3D can significantly and sometimes counter-intuitively affect landscape, with for example rivers on weaker rocks being steeper than on harder rocks because of a combination of factors (dipping strata, $n$, ...). I suggest this manuscript is an excellent candidate for publication in Esurf: (i) the topic is very relevant for the geomorphological community, (ii) the work builds on previous studies (Forte et al., 2018, Perne et al., 2017, Darling et al., 2020, Royden and Perron, 2013) while bringing novelties and (iii) the methods/results support the discussions/conclusions. I have a number of minor points I suggest need a bit of work before publication; most of them are methodological points in order to ensure the robustness of the analysis.

**Specific comments**

First, $k_{sn}$ and $\chi$ are key metrics for this manuscript. They are calculated with $\theta = 0.5$, like a lot of other studies. However, this value is based on Whipple and Tucker (1999), which suggested their range of 0.35 - 0.6 based on a number of assumptions about hydraulic conditions, spatial variations in erosional

processes or uplift/erodibility conditions, and limited number of empirical observations. I suggest that in the case of this study, it is not clear whether these assumptions can be made: for example, different rock types can commonly induce a difference in erosional processes. Recent studies (e.g. Perron and Royden, 2013 or Mudd et al., 2018) demonstrate that 0.5 is not ubiquitous. Then, even within the restricted range of $0.35 < \theta < 0.6$, $k_{sn}$ may vary significantly and in a non-linearly (see Mudd et al., 2018 or this preprint https://doi.org/10.1002/essoar.10505724.1). This can have a strong impact on $k_{sn}$, especially when extracted from real landscapes. I am not suggesting the whole study needs to consider varying $\theta$, but a sensitivity analysis on a specific case might be good to at least show that it does not significantly affect the end results. This is especially true as different values of $n$ are utilised which would mean, as $\theta = m/n$ within the stream power referential, that the area exponent adjusts itself to $n$?

Then, a more minor point is the calculation of $C_H$ involving binning of area data which can be sensitive to the number of bins utilised (see Perron and Royden (2013) or Mudd et al.(2018) figure 2). Along the same lines of comments, I would suggest to add a quick sensitivity analysis on the latter to ensure the consistency of the method.

To finish about the methodological points, I would recommend to add few details (briefly and potentially in the supplemental) about the topographic analysis: which algorithms/methods have been used to calculate slope, drainage area and $k_{sn}$ as these can impact the final data (see Wobus et al. (2006), Mudd et al. (2014,2018) or Gailleton et al (2019) for a few examples - Note that I am not necessarily suggesting the addition of these references).

In term of the manuscript structure and readability, I echo the comment from Reviewer #1: the manuscript is quite long. I do not necessarily mind, but I see how a reasonable shortening or restructuring could benefit interested readers. For example, I find the introduction rather long and it could be split (for example with a "Motivations" subsection starting on line 40). At smaller scales, the paragraph starting on line 68 is relatively long while Figure 2 already says a lot of it in a clear way. The paragraph starting on l. 97 is quite dense and could be shortened and clarified to concisely state the manuscripts aims. Subsections 2.2 and 2.3 could also be summarised while migrating some details to the supplemental materials. The results sections have some repetitions between the description of the figures in the text, their captions and the figure itself.

I noticed different terms used for referring to $k_{sn}$ (e.g. stream steepness, channel steepness, river steepness). I suggest the manuscript would be clearer if this were homogenised. I believe the right term would be *normalised channel steepness* as the metric is $k_{sn}$ and not the not-normalised to a $\theta_{ref}$ $k_s$.

I have a small concern about the field site utilised. The area is quite arid and displays a lot of plateaus. Although the aridity is a nice feature which helps

detect the contact between formations, I wonder if these factors complicate (i) the calculation of drainage area and (ii) the implicit link between area and discharge required by $k_{sn}$ (e.g. Flint (1974))? Maybe this has been studied for the area.

Finally I would raise an opening point. The limitations of this approach are clearly stated in the discussion: the model assumes a detachment-limited law, given erodibility contrasts between layers and discrete values of $n$. I believe the study would really benefit from a subsection in the discussion exploring the impact of more variability in these parameters. In particular, I am thinking about the impact of regularity in the results: would the relationship be that clear with "randomly" varying layer thickness and/or $K$? how fast would the signal get obscured by randomise variations? I acknowledge that I do not know how much work this last point might require and do not consider it as a major missing point, but rather as a valuable potential addition to the contribution.

**line-by-line comments**

*l. 17*: I don't mind the term "stream steepness"; however I would recommend using "channel steepness" as a more common alternative.

*l. 30*: I would replace "and" by "or" as one could suggest stream power has been used in many other situations (all the chi-related works expressed in the stream-power referential following Perron and Royden (2013) as one example among many).

*l. 46*: the work of Lavarini et al. (2019, https://doi.org/10.1029/2018JF004610) also is a nice example of the consequences differential lithology can have on detrital analysis, beyond the sole difference in erosion rates.

*l. 52-56*: I apologise for the inelegant suggestion, but I think this preprint (https://doi.org/10.1002/essoar.10505201.1) would be a relevant reference for the use of channel steepness to explore lithological variations and their implications in landscapes evolution (it is in the final stages of the peer-review process and I hope will be in an accepted form for the authors' revisions).

*l. 65*: Briefly add a couple of words to detail the $k_{sn}$ extraction method (i.e. from S–A, $dz/d\chi$ regression or else).

*Figure 1*: The unit of $k_{sn}$ is $m^{2\theta_{ref}}$ (where $\theta_{ref} = m/n$ within the stream power law referential), also the value of $\theta_{ref}$ needs to be reported.

*l. 75*: Alternatively, studies using steepness to unravel landscape evolution could also misinterpret variations in channel steepness due to lithologic variations as erosion contrasts (e.g. knickzones) due to base level falls. The different set ups in Figure 2 could lead to different type of misinterpretations.

*l. 97*: "Here" ? Do the authors mean "in this contribution"?

*Figure 2*: Nice figure!

*l. 124*: "nonzero" should be 'non-zero"

*l. 130*: "upwind", do you mean you are calculating $dz/dx$ in the upstream direction or with an explicit Euler scheme? Calculating slope in the upstream direction could have some numerical consequences (see Campfort and Govers (2015), https://doi.org/10.1002/2014JF003376).
*l. 136*: see my main comment about $\theta$.

*l. 164*: I don't find it confusing, it makes sense to me!

*l. 180*: Does $\chi_{sp}$ vary within the slope patch? I guess it does not matter here.

*l. 201*: $ka$ refers to relative time (10 ka = 10,000 years ago), I would suggest to stick with *kyrs* and *yrs* for the whole paper.

*l. 202*: This is a rather short time step. Any particular reason?

*l. 291*: Just to make sure, steeper reaches = higher $k_{sn}$ or higher S?
*l. 294*: It is also important to state that the non-linearity of the relationship increases with $|n - 1|$

*l. 300*: I feel like it could be stated more clearly that $n = 1$ is not numerically stable/representable with this equation.

*l. 308*: Why would $C_{HW}$ and $C_{HS}$ be equal?

*l. 330*: replace "you" by "one" or "we".

*l. 345*: It also assumes constant erodibility and layer thickness for each rock type?

*l. 452*: It is not clear why these specific values of $n$ are used.

*Figure 9*: The figure is difficult to read, especially the legends. Again, I wonder how sensitive the data is to the way A is binned.

*Figure 10*: the scatter plots are quite dense and difficult to read. Maybe smaller points, or unfilled symbols or another type of visualisation would make

it clearer?

*l. 780*: Generally accurate for numerically "perfect" data, I suggest it is important to note this.

*Figures S1, S2, S3 and S4*: These figures are very difficult to read, I would really recommend to rethink their style. I am not sure one can extract relevant information from them.

---

## Author Response (AR1)

**Dr. Boulton,**

**Thank you for your thoughtful review and constructive feedback. We address your specific comments below, but the most significant changes to the manuscript include (1) moving material from the methods section into the supplement and (2) cutting the description of figures in the results section. Our intention was to thoroughly address the topics addressed in this study, but we agree that focusing more on brevity has improved the manuscript.**

Clarity and readability.

Unfortunately, I find that the paper is rather long and quite wordy and in places repetitive, so much so that often the key points that the you are trying to are lost on me. My main recommendation would be for you to cut the length of the methods and results sections to make these parts shorter and clearer. The methods section is 11 pages long. Could some of this information go in the supplements for the interested reader but for non-modellers only the key parameters and assumptions are described?

**Response: Thank you for the suggestions. We have moved most of the material from sections 2.2 (Defining the range of erodibility values) and 2.3 (Recording contact migration rates) into the supplement. Those sections focused on considerations that are likely of interest to modelers (e.g., the damping length scale). We also shortened figure descriptions in the results section. We hope these changes have improved the article.**

In addition, in the results section figures are often described in the text using virtually the same words as the accompanying figure captions but the key data, trend or observation that the reader should take away from these plots is not clear. This occurs for example on lines 552- 560, lines 682 – 686; line 704 – 709 etc. This means that often I am confused as to the key point being made and I would prefer the result to be stated not a figure description (e.g., What is high/low r2 values? Lines674/675; what the different spatial patterns? line 600)

**Response: We agree that there was redundancy between the text and captions, so we cut much of the redundant text from figure captions. We also added text to make the key trends/observations more explicit. For example, when introducing Figure 6c, we now clarify that its purpose is to test if equations from Perne et al. (2017) still apply when there are three rock types instead of two. For the lines that were 682-686, I now clarify that this paragraph discusses kinematic wave speed estimates made with our approach. When introducing Figure 10, we now clarify that the purpose is to test the accuracy of the two equations for kinematic wave speed (Equations 12 and 13). To focus more on the key results, we also cut much of the description of Figure 10. We added the $R^2$ values to the lines that were formerly 674-675. For the line that was formerly 600, we added a brief description of how the spatial patterns in erosion rate are different.**

Rock strength

There have been a number of recent papers based on field measurements of rock strength to determine K (i.e., Kent et al., 2020; Zondervan et al., 2020a; b in addition to those studies that you already cite) but you don't refer to these when discussing how you chose the K values for the strong and weak rocks. This is important as several of these studies indicate the in the real world erobilities are many orders of magnitude less than values used in models including these used on here. Given that the stated erodibilites are also stated to 3 significant figures – how did you come to these numbers and what are the implications for your models if all the rocks are 'weak' in comparison to the limited field data available?

**Response: The erodibilities presented in Kent et al. (2020) and Zondervan et al. (2020) have different dimensions relative to the erodibilities we use (m s$^2$ kg$^{-1}$ in those studies vs m$^{0.33}$ yr$^{-1}$ or m$^{-0.5}$ yr$^{-1}$ in this study). Because of these differences, the values cannot be directly compared. The erodibilities we use in this study are comparable to those with the same dimensions reported by Armstrong et al. (2021), however. Our intention was to objectively and systematically explore a large range of erodibilities. We did so by calculating K using prescribed contrasts in slope patch migration rates between strong and week layers (e.g., the strong K value that would produce slope patch migration rates that are 50% of those for the reference weak K and rock-uplift rate U; this approach is described in Section S2). We did not intend to base the erodibilities in our simulations after any particular values reported in the literature in part because of the difficulty in comparing K values when there are differences in dimensions. Because (1) we use these simulations to explore general behaviors and the accuracies of Equations 7, 8, 10, 12, and 13 and (2) the observed behaviors are generally consistent regardless of the reference weak erodibility used (e.g., the different symbol sizes in Figure 10 represent different weak erodibilities and all follow a 1:1 relationship with contact migration rates), we would argue that the erodibilities do not need to be based on specific values reported in the literature.**

Tank Wash

I would like to see greater justification for the use of the Tank Wash site as your 'field' area. It appears that there is no published geological map or other field constraints for the area – so is this really the best location? Although you recognise that this is 'far from ideal' there must be reasons you chose this site over somewhere else nearby on the map? But I'm not clear as to what these are? You do mention the stepped river profile but surely there are other similar locations that have better constrained bedrock data?

**Response: Yes, we agree that some more information can be included for our choice of Tank Wash. We chose Tank Wash for a number of reasons. (1) Regional geologic maps suggest the strata are gently dipping to the west and Tank Wash flows to the east. This orientation limits the complications that might arise in a river with a flow direction that is oblique to the dip direction (or has changes in flow direction relative to unit strike). (2) The lack of vegetation in this area allows us to compare river morphology changes with rock types changes due to the clear color changes that occur in this suite of sedimentary rocks. While a more detailed geologic map might add more certainty on our inferred contacts, the clarity and geometric set up of the Tank Wash natural experiment led us to focus on this particular channel as an example application of the theory and modeling performed in this study. We modified the text in Section 4.4 to clarify our choice of Tank Wash.**

Additionally, what is the justification of using K values across such a large range? Is this using Stock and Montgomery (1999)?

**Response: Thank you for bringing this up. As the reviewer is aware, there is no consensus or theoretical framework on how K should vary in natural systems, thus we purposely explore a broad range of K. We began by selecting three reference weak erodibilities for $n = 1$: $5 \times 10^{-7}$, $10^{-6}$, and $2 \times 10^{-6}$ $yr^{-1}$ (discussed in Section S2). At that point, we had not yet decided to exclude simulations with $n = 1$. We based those reference weak K values for $n = 1$ on previous work (e.g., Figure 11 in Armstrong et al. (2021) and Figures 4a-b and 5a-b in Mitchell and Yanites, 2019). We set the three reference weak erodibilities for $n = 0.67$ and $n = 1.5$ as those that would produce the same slope patch migration rates as the K values for $n = 1$ (for a rock-uplift rate of 0.15 mm $yr^{-1}$). The K values we arrived at through this process are comparable to the K values (when dimensions agree) Armstrong et al. (2021) found by calibrating incision models to rivers upstream of a normal fault in southwestern Montana, USA. The K values for $n = 1.5$ are lower than those for $n = 0.67$, but these K values have different dimensions ($m^{0.33}$ $yr^{-1}$ vs $m^{-0.5}$ $yr^{-1}$).**

On line 845 you state that dating of terraces is used to constrain uplift, but incision recorded by terraces often does not equal uplift owing to the processes of aggradation as well as incision that occur during terrace formation. Do you have any other constraints on uplift? What is the error on this parameter?

**Response: We agree, the incision rates from terraces are not necessarily representative of rock-uplift rates (or relative base level fall rates), but they provide a broad enough constraint to put a range of quantitative values on the analysis of Tank Wash. We recognize, however, the uncertainty associated with making this assumption. As such, we have expanded the discussion of using the incision rates in section 2.6. Importantly, we also clarified in section 3.4 that the erodibilities estimated with our approach are proportionate to the assumed rock-uplift rates. So even if the true base level fall rates are 50% of the rates we use, making that adjustment should produce erodibilities that are 50% of the values we estimate. We realize that our use of Hanksville as a potential real-world example involves many such caveats, but our intent is only to demonstrate how this approach can be applied to a real stream. A study making more specific assertions regarding the properties and history of Tank Wash would require more data than are currently available.**

Minor comments

Line 56 – Reference needed for metrics of rock strength and channel steepness, maybe Zondervan et al. (2020b) or Bernard et al. (2019) would be suitable here.

**Response: Thank you, we now include those references.**

Line 71 – is the word possible in this sentence appropriate – common would be my experience in most regions of the world.

**Response: We agree – our intention was to state that one could consider more complicated examples, not that more complicated scenarios are less likely to be found in nature. We now state that more complicated scenarios are likely in nature.**

Line 208 – what observations are you referring to here?

**Response: When I first began exploring these models, I noticed that some simulations with n = 0.67 would have elevations that gradually increase over time. Conversely, simulations with n = 1.5 would have elevations gradually decrease with time. That characterization is not universal, however, so I have rephrased that section. I simply chose to initialize the stream profiles with the weak layer's expected steepness ((U / K_weak) ^ (1 / n)) when n > 1 and the strong layer's expected steepness ((U / K_strong) ^ (1 / n)) when n < 1. Although the rivers must adjust from these initial conditions, all simulations were fully adjusted within the initialization periods used for our models. I have restructured Figures S1-S4 to better demonstrate both (1) the initial adjustments required for each simulation and (2) that all simulations were fully adjusted within the initialization periods.**

Lines 570 – 585 I'm not sure if this section is describing results or background information, maybe consider the location of this information.

**Response: Thank you for your feedback. We did consider moving these observations. Presenting these observations requires showing the results of our simulations (Figures 7 and 8), however, and we wanted to show those figures in the results section. We have now rephrased that section to make it fit better within the results section.**

Lines 434/437 - Why are dips expressed as negatives?

**Response: When contacts dip in the upstream direction, we express the dip as negative (i.e., contact elevations decreasing in the positive x direction). We use this convention in accordance with studies like Wolpert and Forte (2021).**

References not in pre-print

Bernard T, Sinclair HD, Gailleton B, Mudd SM, and Ford M., 2019. Lithological control on the post-orogenic topography and erosion history of the Pyrenees. Earth Planet Science Letters, v. 518, p. 53–66.

Kent, E., Whittaker, A.C., Boulton, S.J. and Alçiçek, M.C., 2020. Quantifying the competing influences of lithology and throw rate on bedrock river incision. GSA Bulletin.

Zondervan,, J.R., Whittaker, A.C., Bell,. R.E., Watkins, S.E., Brooke,. S.A.S. and Hann, M.G., 2020a. New constraints on bedrock erodibility and landscape response times upstream of an active fault, Geomorphology, v. 351, p. 106937-106937

Zondervan, J.R., Stokes, M., Boulton, S.J., Telfer, M.W. and Mather, A.E., 2020b. Rock strength and structural controls on fluvial erodibility: Implications for drainage divide mobility in a collisional mountain belt. Earth and Planetary Science Letters, 538, p.116221.

**References for Responses:**

**Armstrong, I. P., Yanites, B. J., Mitchell, N., DeLisle, C., & Douglas, B. J. (2021). Quantifying Normal Fault Evolution from River Profile Analysis in the Northern Basin and Range Province, Southwest Montana, USA. *Lithosphere*, *2021*(1), 7866219.**

**Mitchell, N. A., & Yanites, B. J. (2019). Spatially variable increase in rock uplift in the northern US Cordillera recorded in the distribution of river knickpoints and incision depths. *Journal of Geophysical Research: Earth Surface*, *124*(5), 1238-1260.**

**Wolpert, J. A., & Forte, A. M. (2021). Response of Transient Rock Uplift and Base Level Knickpoints to Erosional Efficiency Contrasts in Bedrock Streams. *Earth Surface Processes and Landforms*.**

Dr. Gailleton,

Thank you for your thoughtful review and constructive feedback. First, we will address the influence of different m/n values. We have added twelve additional simulations in which m/n values are varied between 0.3, 0.5, and 0.7. The results from those simulations are shown in the supplement but summarized in the main text. Overall, different m/n values do not seem to substantially change the dynamics we study when contacts are horizontal. Although erodibilities cannot be directly compared when drainage area exponent m values differ, these new simulations have the same $K^*$ value ($K^* = 9.50$ for the three simulations n = 0.67 simulations and $K^* = 0.125$ for the three simulations with n = 1.5). Because the simulations have the same $K^*$ value, the erosion rates in weak and strong rock types are about the same.

When contact dips are non-zero, however, river behavior depends more strongly on m/n. For example, in Figure 7c (a simulation with n = 0.67) we emphasized that peaks in erosion rate within the weak unit increase in magnitude with drainage area. The new simulations we present demonstrate that the rate of change in the magnitudes of erosion rate peaks depends on m/n. For example, when m/n = 0.3 the erosion rate peaks have a smaller range (e.g., from about 6E/U to about 17E/U in Figure S12a). When m/n = 0.7, the erosion rate peaks have a large range (from about 4E/U to about 26E/U in Figure S12c). Although Figure S16 shows that contact migration rates are still well represented by Equations 12 and 13 (formerly Equations 14 and 15), the influence of m/n on the covariation of erosion rates with drainage area is an important consideration for non-zero contact dips. For example, because m/n values influence the covariation of erosion rates and drainage area they may also influence both spatial contrasts in erosion rate and drainage reorganization. We now discuss these results in sections 3.3.1, 3.3.3, and 4.2.

We decided to bin results by drainage area for visual clarity in our figures (i.e., instead of having a dense cloud of all measured contact migration rates in Figures 4 and 9). To address the influence of our binning approach, we have added a new figure to the supplement. This figure is a version of Figure 9 with 20 drainage area bins instead of 10. The contact migration rates and estimated kinematic wave speeds are generally the same, but there are slight differences (e.g., the $R^2$ value in subplot (b) changes from 0.37 to 0.41). We now point out this consideration in the main text (Section 3.3.3).

To improve the readability of the article, we have cut material from the main text (especially from figure captions) and moved some material to the supplement (e.g., much of sections 2.2 and 2.3). Our intention was to thoroughly address the topics discussed in the manuscript, but we agree that focusing on brevity has improved the article. We also added a section to the supplement detailing how we extracted and processed channel profile data (Section S1). As you suggested, we also (1) added a new subsection called "Motivations" in the introduction (Section 1.1), (2) shortened the description of Figure 2, and (3) homogenized our use of the term "channel steepness" (rather than stream steepness).

We appreciate your feedback regarding the scaling between drainage area and discharge in the area of Hanksville, UT. Unfortunately, we are not aware of any such work in the area that focuses on the broad timescales (i.e., geomorphically significant flows) pertinent to the stream power model. We have expanded our discussion of using Tank Wash as a potential real-world example in section 4.4. We emphasize that our intention is only to demonstrate how the methods developed in this study could be applied to a real stream. A detailed study of Tank Wash that makes more specific assertions regarding its properties (e.g., erosion rates) and history would require more data than are currently available (e.g., contact locations and dips).

We have expanded the discussion of units with randomly varying erodibilities and layer thicknesses (Section 4.2, in the paragraph discussing how rock strength contrasts can drive landscape transience). Although the simulations in scenario 2 only used three rock types instead of two, this scenario was meant to provide insight into how channel slopes would vary with any number of different units. Specifically, scenario 2 shows that the channel slopes and erosion rates in each unit will adjust to allow for a consistent trend in kinematic wave speed across the profile. The exact channel slopes and erosion rates required depend on the distribution of erodibilities among the exposed units. In Section 4.2, we now emphasize that the exposure of a much stronger unit could (1) lower the kinematic wave speeds across the profile and (2) alter the erosion rates in other units. River incision through units with widely varying erodibilities and thicknesses might cause streams to be in a constant state of adjustment, preventing them from truly achieving a dynamic equilibrium.

Our responses to your comments below are shown in bold text.

**line-by-line comments**

l. 17: I don't mind the term "stream steepness"; however I would recommend using "channel steepness" as a more common alternative.

**Response: We have replaced "stream steepness" with "channel steepness" throughout the article.**

l. 30: I would replace "and" by "or" as one could suggest stream power has been used in many other situations (all the chi-related works expressed in the stream-power referential following Perron and Royden (2013) as one example among many).

**Response: We did not intend to suggest these examples portray all uses of the stream power model. To clarify that there are many other implementations of the stream power model, we now begin this list with the phrase "… the stream power model has been used in many applications including: (1) …"**

l. 46: the work of Lavarini et al. (2019, https://doi.org/10.1029/2018JF004610) also is a nice example of the consequences differential lithology can have on detrital analysis, beyond the sole difference in erosion rates.

**Response: Thank you for the suggestion, we have added a sentence discussing that reference.**

l. 52-56: I apologise for the inelegant suggestion, but I think this preprint (https://doi.org/10.1002/essoar.10505201.1) would be a relevant reference for the use of channel steepness to explore lithological variations and their implications in landscapes evolution (it is in the final stages of the peer-review process and I hope will be in an accepted form for the authors' revisions).

**Response: Thank you for the suggestion, that article is certainly pertinent to this work. That article does not seem to be published yet, but we expect it will be soon. At that point, we will include the reference in this article.**

l. 65: Briefly add a couple of words to detail the $k_{sn}$ extraction method (i.e. from S-A, $dz/d\chi$ regression or else).

**Response: We have added a new section to the supplement detailing how we extracted channel steepness (Section S1).**

Figure 1: The unit of $k_{sn}$ is $m^{2\theta ref}$(where $\theta_{ref}$ = m/n within the stream power law referential), also the value of $\theta_{ref}$ needs to be reported.

**Response: We displayed the units of steepness there as meters because we are using a reference concavity of 0.5. We now clarify that the $k_{sn}$ values in Figure 1 use a reference concavity of 0.5.**

l. 75: Alternatively, studies using steepness to unravel landscape evolution could also misinterpret variations in channel steepness due to lithologic variations as erosion contrasts (e.g. knickzones) due to base level falls. The different set ups in Figure 2 could lead to different type of misinterpretations.

**Response: We agree and have added a sentence emphasizing the potential for such misinterpretations.**

l. 97: "Here" ? Do the authors mean "in this contribution"?

**Response: Yes – we now use the phrase "in this contribution."**

Figure 2: Nice figure!

**Response: Thank you!**

l. 124: "nonzero" should be `non-zero"

**Response: We have made that change throughout the manuscript.**

l. 130: "upwind", do you mean you are calculating dz=dx in the upstream direction or with an explicit Euler scheme? Calculating slope in the upstream direction could have some numerical consequences (see Campfort and Govers (2015), https://doi.org/10.1002/2014JF003376).

**Response: We use "upwind" here to refer to the downstream direction (as used in Royden and Perron (2013)). We are not calculating slopes in the upstream direction.**

l. 136: see my main comment about θ.

**Response: Thank you for your feedback. To address the role of different m/n values, we have added new simulations to the supplement (as discussed above).**

l. 164: I don't find it confusing, it makes sense to me!

**Response: I'm glad to hear that!**

l. 180: Does $\chi_{SP}$ vary within the slope patch? I guess it does not matter here.

**Response: Yes, there are many slope patches each with their own $\chi$ values. The dynamics of these rivers (i.e., spatial variations in erosion rate due to contact migration) would likely complicate the dynamics of slope patch migration rates, however. We don't want to focus on those details further in this article as such details would detract from the focus of the study.**

l. 201: ka refers to relative time (10 ka = 10,000 years ago), I would suggest to stick with kyrs and yrs for the whole paper.

**Response: Yes, I normally use that convention. When I was preparing this article, however, I remember reading author instructions saying to always use "a" for years. I cannot find those instructions right now, so it is possible my memory is inaccurate. I now use "yrs" throughout the article.**

l. 202: This is a rather short time step. Any particular reason?

**Response: When erodibility contrasts are high, erosion rates can become very high (e.g., E > 10 U). We used a small time step to avoid numerical instabilities.**

l. 291: Just to make sure, steeper reaches = higher $k_{sn}$ or higher S?

**Response: Higher slopes will lead to higher steepness values, although drainage area and m/n must be considered of course.**

l. 294: It is also important to state that the non-linearity of the relationship increases with |n - 1|

**Response: Thank you, we have added that clarification.**

l. 300: I feel like it could be stated more clearly that n = 1 is not numerically stable/representable with this equation.

**Response: Thank you, we now emphasize that point.**

l. 308: Why would $C_{HW}$ and $C_{HS}$ be equal?

**Response: It is my understanding that in his review for Perne et al. (2017), Kelin Whipple suggested they should focus on the potential for equal kinematic wave speeds in strong and weak units. Darling et al. (2020) also focused on equal $C_{HW}$ and $C_{HS}$ (with Whipple being a coauthor in that article). Proceeding with the assumption that $C_{HW}$ and $C_{HS}$ are equal in these numerical models can be motivated either through (1) observing the spatial variations in erosion rate that occur in these models (e.g., Figure 3) or (2) geometric reasoning (e.g., Darling et al., 2020).**

l. 330: replace "you" by "one" or "we".

**Response: Thank you, we have made that change.**

l. 345: It also assumes constant erodibility and layer thickness for each rock type?

**Response: Yes, it does. Temporal changes in the erodibilities and/or thicknesses of uplifted units would cause gradual adjustments in fluvial relief. We have expanded the discussion of these issues in Section 4.2. Such adjustments would generally be very gradual, however (i.e., the timescale to uplift the unit with a new erodibility or thickness across the profile). As discussed above, if a new rock type with a substantially higher or lower erodibility was exposed this new unit could cause an increase or decrease in the kinematic wave speeds across the profile, respectively (i.e., the moderate trend in $C_H$ maintained across the profile would shift due to the new erodibility). Such a change would alter the erosion rates in each layer.**

l. 452: It is not clear why these specific values of n are used.

**Response: We now emphasize that a wide range of n values are possible, but our intention is to provide a small selection of examples with n values less than or greater than one.**

Figure 9: The figure is difficult to read, especially the legends. Again, I wonder how sensitive the data is to the way A is binned.

**Response: We have reformatted Figure 9 and hope it is easier to read now. We also added a version of Figure 9 to the supplement with 20 drainage area bins instead of 10 (Figure S17). This example demonstrates that our binning approach does have a slight impact on our results (e.g., the $R^2$ value for subplot (b) changes from**

**0.37 to 0.41). The binning approach was mainly used to improve the visual clarity of our graphs (i.e., instead of a dense cloud including all measurements of contact migration rates).**

Figure 10: the scatter plots are quite dense and difficult to read. Maybe smaller points, or unfilled symbols or another type of visualisation would make it clearer?

**Response: Thank you for the feedback. Our intention was not for the reader to extract insight regarding specific points within the scatter plots. Instead, we only intended for the reader to see that all points follow a 1:1 relationship between measured contact migration rates and estimated kinematic wave speed. There is some scatter, however, and we used the dashed lines to provide context for the magnitudes of such deviations (e.g., the measured and estimated data are always within a factor of 2 of each other).**

l. 780: Generally accurate for numerically "perfect" data, I suggest it is important to note this.

**Response: Thank you for the suggestion, we have added that clarification.**

Figures S1, S2, S3 and S4: These figures are very difficult to read, I would really recommend to rethink their style. I am not sure one can extract relevant information from them.

**Response: Thank you for your feedback. We have reformatted these figures to better convey our intended message. During the beginning of the simulations, the streams need to adjust from the initial conditions. The maximum elevations can gradually increase or decrease during this adjustment. The adjustment time is dependent on both the initial conditions and the rock-uplift rate (i.e., the time required for a contact to be uplifted across the fluvial relief). The final maximum elevation is not always obvious before the simulation has been run, however, as spatial variations in erosion rate can complicate that consideration (especially when dips are non-zero). The purpose of Figures S1-S4 is to show that we gave the streams enough time to adjust from the initial conditions. Figures S1c, S2c, S3c, and S4c all have one line for each simulation used – there is a lot going on in each subplot, but the important observation is that all simulations have a relatively narrow range in elevations (e.g., always within ~10% of the final maximum elevation, rather than the large changes in elevation that can occur during the initial adjustment). We have expanded the discussion of these issues in section 2.3.**